# Robust and Actively Secure
# Serverless Collaborative Learning

**Olive Franzese**[*†1], **Adam Dziedzic**[*‡5], **Christopher A. Choquette-Choo**[3],
**Mark R. Thomas**[2], **Muhammad Ahmad Kaleem**[2], **Stephan Rabanser**[2], **Congyu Fang**[2]
**Somesh Jha**[3,4], **Nicolas Papernot**[2], **Xiao Wang**[1]
[1]Northwestern University, [2]University of Toronto and Vector Institute, [3]Google,
[4]University of Wisconsin-Madison, [5]CISPA

## Abstract

Collaborative machine learning (ML) is widely used to enable institutions to learn better models from distributed data. While collaborative approaches to learning intuitively protect user data, they remain vulnerable to either the server, the clients, or both, deviating from the protocol. Indeed, because the protocol is asymmetric, a malicious server can abuse its power to reconstruct client data points. Conversely, malicious clients can corrupt learning with malicious updates. Thus, both clients and servers require a guarantee when the other cannot be trusted to fully cooperate. In this work, we propose a peer-to-peer (P2P) learning scheme that is secure against malicious servers and robust to malicious clients. Our core contribution is a generic framework that transforms any (compatible) algorithm for robust aggregation of model updates to the setting where servers and clients can act maliciously. Finally, we demonstrate the computational efficiency of our approach even with 1-million parameter models trained by 100s of peers on standard datasets.

## 1   Introduction

To leverage data that is located across different clients, service providers increasingly resort to collaborative forms of distributed machine learning. Rather than centralize the data on a single *server*, data remains on the owner's (client's) device(s), which could be a consumer's phone or bank/hospital's local data center. Take the canonical example of federated learning (FL) [34]. Rather than share data, clients instead send model updates to the server. *Our work caters to settings where neither clients nor servers can be entirely trusted to faithfully participate in the Collaborative Learning (CL) protocol.*

For example, consider if a group of banks wished to learn a better fraud detection model. Banks may not be able to directly share data [15] and further because banking is a competitive industry, it must be assumed that banks will deviate from the protocol if it serves their interest. On one hand, malicious server banks may breach the intuitive confidentiality of CL. A long line of work [9, 10, 24, 37, 43, 49, 50, 52] has shown that when the server acts maliciously, it can, for instance, construct model parameter values that exactly extract client data from (even aggregated) model updates. To protect client data from servers acting maliciously, it is thus paramount to design approaches to CL where no single server can have full control over the orchestration of the protocol. On the other hand, malicious client banks may entirely prevent learning by submitting poor updates. This may be intentional as in model poisoning attacks [2, 6, 45, 46] or unintentional if their dataset contained malformed data. Though a separate line of work [25, 26, 29, 30, 32, 38, 48] has studied how to robustly learn in the face of malicious updates (or data), there are none that have studied how to integrate such robust learning algorithms within a protocol that is secure to malicious servers. *In this work, we design the first scheme that is robust to the harms of both malicious server(s) and clients, which are shown in Figure 1.*

---

[*]Equal Contribution.

[†]Correspondence to: nicholasfranzese2026@u.northwestern.edu and adam.dziedzic@cispa.de

[‡]Project Lead. Work done while the author was at the University of Toronto and Vector Institute.

We observe that asymmetric power is the fundamental requirement for malicious servers to breach user data privacy. Thus, we design a fully-decentralized peer-to-peer (P2P) learning protocol where each participant (e.g., bank), or *peer* herein, can equally contribute to the role of the server aggregating updates (and of a client computing updates). Further, we ensure that no single peer has the power to orchestrate the protocol—instead, we elect a committee of peers to perform the aggregation at any given training round in a way that requires no central or trusted third party (see Section 3 for the full threat model). On the other hand, there is now a greater need for protection against malicious clients as the distributed nature may increase

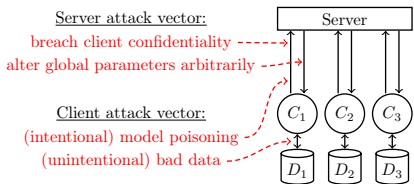

Figure 1: **Motivation for P2P Learning**. Current collaborative learning approaches are vulnerable to both client (denoted as $C$ with data $D$) and server attack vectors. Our framework tackles all of these vulnerabilities simultaneously.

the chances of intentional poisoning or bad data quality interfering with learning (*e.g.,* due to fewer resources among some banks and/or competitive advantages). Thus, we ensure that our protocol can efficiently integrate with classical approaches for robustness against malicious clients, such as RSA [32], FL Trust (FLT) [14], or Centered Clipping (CC) [29]. Importantly, our work generalizes the setups of these works and introduces the general framework that adapts any (compatible) algorithm for robust aggregation of model updates to settings where servers and clients may behave maliciously.

To achieve this, our approach builds on cryptographic multi-party computation (MPC) protocols. This allows peers to collectively emulate the server's role while being robust against the collusion of a subset of these peers that may act maliciously. However, naively combining these with (insecure) robust aggregation techniques incurs prohibitive overhead because the server computation for robust aggregation, which must be securely computed in MPC, is almost always of a complexity that leads to a high multiplicative slowdown. We design a framework that modularizes the processing steps of robust aggregation so as to select the most suitable cryptographic building blocks for each one, leading to significant computational improvements. One such improvement is our proposed *computational surjectivity*. We show that aggregation algorithms with component functions satisfying this property can efficiently obtain security while still guaranteeing robustness against malicious peers; we also show that existing robustness algorithms satisfy this property, or can be tailored to do so.

To summarize, our contributions are the following:

1. We design the first collaborative learning protocol that operates under the malicious threat model and is robust to both malicious clients and servers. We provide a simulation-based proof of its cryptographic security.

2. We design our protocol as a generic compiler that can convert broad categories of robust aggregation algorithms to our improved security model efficiently. This modular approach enables practitioners obtain rigorous security guarantees while selecting the most appropriate model poisoning defense for their use case. To demonstrate our framework's flexibility, we generate malicious-secure protocols for three existing robust aggregation algorithms. We show empirically that the generated protocols retain their robustness guarantees.

3. We demonstrate the computational efficiency of our protocols. We benchmark our protocols up to 1 million parameter models, and thousands of peers. For example, we show that the aggregation step of our malicious-secure implementation of robust aggregation with RSA [32] obtains a per-round CPU time of roughly 46 seconds with $10^5$ parameters when trained by 1000 peers.

## 2 Related Work

Federated learning is perhaps the most studied collaborative learning framework [28, 35]. Most related to ours are variants based on Secure Aggregation (SecAgg) [11] that provide confidentiality of gradient transmission. However, existing work does not provide robust aggregation within SecAgg and is focused on the single-server setting, or additionally on their use for tighter differential privacy guarantees [16, 27, 41, 47]. In contrast, we focus solely on confidentiality in the distributed server setting with robust aggregation. Other works include CaPC [17] but this requires a trusted third party to reduce the computational overhead. We make no such assumptions. In Swarm Peer-2-Peer learning [44], participants can dynamically join or leave the collaboration and are enrolled via a Blockchain smart contract. There is no central party and each per-round server is dynamically elected

| | Property | Update Confidentiality | Malicious Clients | Malicious Server | Aggregation Committee | Robust Aggregation |
|---|---|---|---|---|---|---|
| Method | Prevented Attacks | Plaintext Inspection | Poisoning or Backdooring [2, 6, 45, 46] | Gradient Inversion [23, 50, 37] [43, 49, 50] | Data Reconstruction [9, 10] or Degrade Utility | Malformed Data |
| SecAgg v1 [11] | | ✓ | ✗ | ✗ | ✗ | ✗ |
| SecAgg v2 [5] | | ✓ | ✗ | ✗ | ✗ | ✗ |
| CaPC [17] | | ✓ | ✗ | ✗ | ✓ | ✗ |
| Swarm P2P Learning [44] | | ✗ | ✗ | ✗ | ✓ | ✗ |
| Biscotti [39] | | ✓ | ✗ | ✗ | ✓ | ** |
| Eiffel MS [20] | | ✓ | ✓ | ✓ | ✗ | * |
| Acorn MS [4] | | ✓ | ✓ | ✓ | ✗ | * |
| RS-P2P SHS (Ours) | | ✓ | ✗ | ✗ | ✓ | ✓ |
| RS-P2P MS (Ours) | | ✓ | ✓ | ✓ | ✓ | ✓ |

Table 1: **Comparison of Security Models between Aggregation Protocols.** Robust aggregation provides protection against data poisoning by clients in the collaboration protocol. Update confidentiality guarantees that an individual updated from a client is not revealed. SHS denotes Semi-Honest Security while MS is Malicious Security. *Guarantees data integregity, not robust aggregation of updates. **Only under a single robust aggregation protocol.

via Blockchain smart contracts. Crucially, Swarm Learning supports neither secure (confidentiality-preserving) nor robust aggregation—it uses standard parameter averaging.

Biscotti [39] incorporates robustness to poisoning by combining Multi-Krum [7] and secure aggregation through Shamir secret-sharing. Its core parts are a verification committee that runs robust update selection, and aggregation committee that computes the final model update. However, Biscotti only guarantees security in the semi-honest setting and is solely compatible with Multi-Krum, which is not always the preferable robustness algorithm [29]. Blockchain is also used as an alternative to the centralized aggregator in FL to deal with malicious participants or servers in [51]. The initial model is uploaded on the blockchain following which the participants train local models, then sign on hashes with their private keys, and upload the locally trained models to the blockchain. The validity of the uploaded models is verified with digital signatures and Multi-Krum. Algorand is used as the consensus algorithm in the blockchain system to update the global model. However, it uses a single leader for each training round and is compatible only with Multi-Krum.

Konstantinov and Lampert [31] present a distributed robust learning procedure that allows for robust learning from untrusted sources. Distributed Robust Learning (DRL) [22] is another approach to robust learning which uses a divide and conquer strategy. However, none of the papers achieves the two notions of robustness at the same time. Closest to our work are those that look to combine data integrity and confidentiality (security) [4, 20]. However, these works are crucially different from ours in that they perform checks on the underlying data of each client, not the update—then, these protocols drop clients with poor data. Because these approaches operate over a different input, they may be used simultaneously with ours.

## 3 Threat Model

Collaborative learning is conducted among a set of parties (herein, *peers*) performing one of two roles: a *client* (or *worker*) who performs learning on a local dataset, or a *server* that aggregates the many client updates. Our protocol differs in two main ways: first, it is conducted among a set of peers (parties) which can perform either role, and second, the role of the server is performed by a subset of peers termed the *aggregation committee*. To align with prior literature, we sometimes refer to peers as clients or servers when they are performing those respective roles. We consider a malicious threat model where clients and servers may perform arbitrary adversarial actions to interfere with the protocol. Malicious behavior in the two roles may include, but is not limited to the following.

1. **Malicious Clients** may attempt to (1) lower the quality of the trained model by sending distorted model updates. This may take the form of both (a) intentional model poisoning attacks, and (b) unintentional problems such as errors in computation, and skewed or incorrect local data sets. They may also attempt to (2) steal information about the other peers' data, i.e. break confidentiality, e.g. by colluding with other malicious peers and sharing transcripts of the protocol execution.

2. **Malicious Servers / Committee Members** may attempt to (1) reconstruct individual data points from the clients' updates, thus breaking data confidentiality, which can be achieved by arbitrarily modifying model parameters or colluding with other parties (Committee Members or Clients), (2) inappropriately change the shared model by e.g. omitting updates from selected clients, adding in bogus updates, or otherwise altering the global model updates.

We compose multiple cryptographic primitives, including secure committee election, verified secret sharing, distributed zero knowledge proofs, and secure multiparty computation. The assumptions and guarantees of the individual primitives are in Appendix B.2. *Importantly, their composition is secure under universal composability* [13]. Our overall protocol operates under the standard assumptions of authenticated point-to-point secure channels between peers and a bounded proportion of adversarial peers (see Appendix B.1 for details). The following are the formal guarantees of our protocol.

- **Correctness of aggregation.** Given a publicly known update aggregation function $F^R$ and that clients submit local updates $x_1, x_2, \ldots, x_n$, the returned global update will be equal to $F^R(x_1, x_2, \ldots, x_n)$. See the following Section 4 for details.

- **Confidentiality of client updates.** During protocol execution, no peers gain information about any other's update $x_i$ beyond what is implicitly revealed by the aggregate result $F^R(x_1, x_2, \ldots, x_n)$.

- **Robustness to poisoning.** An accurate model will be trained even if some subset of clients submit arbitrary poisonous updates. Our framework compiles existing robust aggregation algorithms into a stronger security model. Thus, the details of this guarantee depend on the underlying algorithm.

- **Malicious (active) security.** The above conditions hold even when a subset of parties actively perform arbitrary malicious behavior, including but not limited to: collusion between malicious peers, attempts to deviate from any part of the protocol, and submission of poisonous local updates.

**Problem Setup.** To construct a collaborative learning protocol that is robust against both malicious clients and servers, we must decentralize the task of update aggregation. Accordingly, P2P learning is conducted among a set of *peers*, who may be assigned the role of client or server.

## 4 Robust and Actively Secure Framework

Our framework efficiently lifts the robust aggregation algorithms (*e.g.,* the aforementioned RSA, FLT, or CC) to the P2P learning setting with guaranteed malicious-secure (or, actively-secure) protocol fidelity. This security model guarantees both confidentiality and protocol fidelity against peers that may take arbitrary actions to disrupt the P2P learning protocol execution—fidelity is maintained by retaining the model fidelity guarantees of an underlying robust aggregation algorithm. Indeed, we previously mentioned that many algorithms provide model fidelity against poisonous adversaries in the single-server setting [7, 25, 29, 30, 32, 38]. Each algorithm makes different assumptions about the threat model, *e.g.,* how many times a given malicious client can participate, what sort of malicious update they send, what the underlying data distribution is, etc. Thus, rather than pinning our framework on a single robustness algorithm, we propose a modular design that encompasses a broad class of such robust aggregation algorithms designed for the single-server setting.

### 4.1 Framework Design

In order to strengthen the security models of a broad class of robust aggregation algorithms, we design a modular template (Figure 2), which organizes aggregation algorithms in terms of three functions:

$$F^C : \mathcal{D} \times \mathcal{S} \times \Omega \to U \qquad F^P : U \to V \qquad F^R : V^m \to \Omega$$

The first function, $F^C$, represents the computation of client updates based on local data, state, and global model parameters; accordingly, $\mathcal{D}$ is the space of possible client datasets, $\mathcal{S}$ is the space of local states, $\Omega$ is the space of global model parameters, and $U$ is the space of client updates. In the trusted single-server setting, each client computes $F^C$ and sends their update $\boldsymbol{u}_i \in U$ to the server. Next comes the server's computation. We break the server's work into two parts: a preprocessing function $F^P$ and an aggregation function $F^R$. The former transforms each client update to a preprocessed domain $V$, and the latter combines the preprocessed local updates into a global model update $\boldsymbol{w} \in \Omega$. Our primary contribution is the design of a protocol that lifts any robust aggregation algorithm described in terms of these functions to a stronger security model. The security model in question

---

**Trusted Single-Server Robust Aggregation**

**Public Functions:** Single-server robust aggregation algorithms are defined by three functions:
- $F^C(\cdot)$ – client-side update computation
- $F^P(\cdot)$ – server-side update preprocessing
- $F^R(\cdot)$ – server-side update aggregation.

**Input:** Global parameters $w$ from the previous round. Each client $P_i$ has input data; all participants have local state st.

**Client update:**

1. Each client $P_i$ computes $u_i \leftarrow F^C(\mathsf{data}, \mathsf{st}, w)$ and sends update $u_i$ to the server.

**Server preprocessing:**

2. For each $i \in [m]$, the server obtains $v_j$ for all $j \in S_i$ and computes $v_i \leftarrow F^P(\{u_j\}_{j \in S_i})$.

**Server update:**

3. Server computes $w \leftarrow F_R(\{v_i\}_{i \in [m]})$ and sends $w$ to all clients.

---

Figure 2: **Template for single-server robust aggregation.**

is secure against malicious/active clients *without relying on a trusted server*, all while retaining the protection against poisoning attacks offered by the original algorithm.

**Protocol Description.** Peers carrying out a P2P Learning protocol (Figure 3) begin by randomly selecting an aggregation committee, the size of which is parameterized to guarantee an honest majority with all but negligible probability (see Appendix B for details). Since the committee is honest-majority, it can use secure MPC (secure Multi-Party Computation) and VSS (Verifiable Secret Sharing) schemes later in the protocol. All clients then compute local updates via $F^C$ and preprocess those updates via $F^P$. Peers secret share their updates with VSS and pass shares to the aggregation committee. Each member of the committee receives a share of a local update from every peer. The committee uses distributed zero knowledge proofs to ensure that all updates are well-formed outputs of $F^P$—Section 4.2 discusses in detail how to do so with practical efficiency. Finally, $F^R$ is computed by the aggregation committee by using the shares as input to a malicious-secure MPC protocol, and committee members send the resulting global model update to all peers. This protocol relies on users remaining online throughout, an assumption that may not always be practical. In Section 4.3, we discuss how to relax this.

**Strengthened Security Model.** In the single-server setting, the computation of $F^R$ is handled by a single party. This makes it vulnerable to tampering – a malicious server may breach client confidentiality, omit updates from certain clients, modify updates, or simply make arbitrary changes to the global model. Our framework lifts aggregation algorithms to a security model where none of that is possible. Distributing the computation of $F^R$ to an honest-majority committee equipped with malicious-secure MPC means that $F^R$ is computed with guaranteed correctness and that no information about the local updates is leaked in the process. Further, using VSS guarantees that no committee member can breach the confidentiality of client updates before the computation of $F^R$, and that it is (with high probability) impossible to modify client updates before the computation of $F^R$ without being caught. Further, since the committee is majority-honest, all peers can guarantee the received global update is correct by taking the majority result received from the committee members.

**Obtaining Practical Efficiency.** It is possible to strengthen the security model of almost any distributed computation by simply running it inside of a generalized MPC protocol, but doing so usually results in unbearable computational overhead since MPC substantially amplifies the cost of most operations. A key challenge that we surmount is *strengthening security whilst maintaining the efficiency necessary to scale to real-world collaborative learning scenarios*. The design choices we employ while formulating our protocol make this possible. For example, in applications of robust aggregation with a trusted single-server, the role of the server is typically executed by a data center with high compute capabilities. In such a setting it is beneficial to minimize client-side computation and shift the compute responsibility to the server wherever possible.

In contrast, collaborative learning with no trusted parties requires a committee to aggregate client updates, and operations performed in MPC by the committee are especially *costly*. Thus it becomes

---

**Secure P2P Learning Against Malicious and Poisonous Adversaries**

**Protocol:**

1. The clients randomly select an aggregation committee $C \subset \{P_i\}_{i \in [m]}$.

2. Each client $P_i$ applies local computation $\boldsymbol{u}_i \leftarrow F^C(\mathsf{data}, \mathsf{st}, \boldsymbol{w})$.

3. For each client $P_i$, compute $\boldsymbol{v}_i \leftarrow F^P(\boldsymbol{u}_i)$.

4. $P_i$ secret shares $\boldsymbol{v}_i$ to obtain $[\boldsymbol{v}_i]$ and sends one share to each $P_j \in C$.

5. If $F^P$ is not computationally surjective, $P_i$ uses Distributed Zero Knowledge (DZK) to prove to the committee $C$ that $\boldsymbol{v}_i$ is correctly computed from some $\boldsymbol{u}_i$ of $P_i$'s choice. Otherwise, $P_i$ uses DZK to prove that $\boldsymbol{v}_i \in V$.

6. If $\mathsf{Domain}(F^R) \neq \mathsf{Image}(F^P)$, $P_i$ uses DZK to prove to the committee $C$ that $\boldsymbol{v}_i \in \mathsf{Image}(F^P)$.

7. All committee members $P_j \in C$ input shares $[\boldsymbol{v}_i]$ for all $i \in [n]$ to a $|C|$-party computation protocol in order to compute $\boldsymbol{w} \leftarrow F^R(\{\boldsymbol{v}_i\}_{i \in [n]})$. Committee members send $\boldsymbol{w}$ to all clients.

---

Figure 3: **Main protocol outline for the malicious setting.**

beneficial to offload as much of the computation as possible to the client-side. Our template (Figure 2) and protocol (Figure 3) do this by separating the trusted server's work into two parts, $F^P$ and $F^R$, and shifting the work of computing $F^P$ to the *clients*. This dramatically reduces the computational burden of the aggregation committee, but introduces potential concerns about the correctness of the underlying aggregation algorithm. Namely, in the trusted server setting $F^P$ is guaranteed to be computed correctly since it is executed by a trusted party, but a malicious client may introduce arbitrary faults into the computation of $F^P$. To prevent this while maintaining confidentiality, one *could* use a zero-knowledge proof to guarantee that $F^P$ was computed correctly, however this would introduce substantial computational overhead. We achieve a much more efficient result by instead verifying that each peer's local update is *well-formed*—that it properly falls within the preprocessed domain $V$. We observe that if $F^P$ has a certain property, which we call *computational surjectivity*, verifying that the update is within $V$ is just as good as verifying correct computation of $F^P$, even though the former comes at substantially lower cost.

### 4.2 Computational Surjectivity

Our key insight is that by leveraging the properties of robust aggregation, we can relax certain requirements on the correctness of $F^P$. These relaxed requirements allow us to offload computation of $F^P$ to the client-side, while also avoiding the computational overhead of a full zero-knowledge proof that $F^P$ was computed correctly.

A robust aggregation algorithm guarantees that even when adversaries provide arbitrary values as the output of $F^C$, a satisfactory output of $F^R$ will be computed. Accordingly, we observe that as long as *some* valid output of $F^C$ maps to each client's output of $F^P$, the final global update will be computed properly. Thus if $F^P$ is a surjective function (i.e. if $\forall \boldsymbol{v} \in V, \exists \boldsymbol{u} \in U : \boldsymbol{v} = F^P(\boldsymbol{u})$), it is only necessary to verify that $\boldsymbol{v}_i \in V$ for all client updates $\boldsymbol{v}_i$ in order to correctly compute $F^R$. Below we specify a computational analogue of surjectivity—we require the preimage can be found in polynomial time so the whole protocol can achieve simulation security (Appendix B.3 has details).

**Definition 1** *A function $f : U \to V$ is* computationally surjective *if there is a probabilistic polynomial-time algorithm $\mathcal{A} : V \to U$ such that for any $v \in V$, we have $f(\mathcal{A}(v)) = v$.*

In general, we have no guarantees on the structure of $F^P$ and so peers must prove in zero knowledge that $\boldsymbol{v}_i$ is the result of a valid computation of $F^P$ (Figure 3, step 5). But if $F^P$ is computationally surjective, then all possible $\boldsymbol{v}_i \in V$ are implicitly the output of some computation of $F^P$. Thus, it only becomes necessary to prove that the shares of each peers' input reconstructs a point within $V$.

Theorem 1 (proof in Appendix B.3) states the security of this protocol in the malicious setting.

**Theorem 1** *For any single-server robust aggregation algorithm described in $(F^C, F^P, F^R)$ as in Figure 2, the protocol described in Figure 3 is a secure P2P learning protocol against malicious clients and servers when the underlying MPC scheme is secure.*

## 4.3 Tolerating Peers Disconnecting

Peers cannot always be assumed to remain connected throughout an entire protocol execution, e.g., when peers are mobile devices [11]. Further, it is also common to subsample a small portion of peers as clients to avoid high computation/communication costs [47]. We show how to account for both of these practical settings with minimal modifications to our protocol.

**Tolerance to Users Dropping.** Our protocol includes two areas where peers must collaborate on the cryptographic protocol: the (client) work of computing updates and the (server / committee) work of aggregating updates. Our protocol already gracefully tolerates any number of clients dropping so long as the the pool of remaining clients meets the assumptions of the underlying robust aggregation algorithm. In this case our protocol's output would be just as if those peers did not participate. Our protocol can also tolerate committee member dropout *with no impact on the output of the protocol* by proportionally increasing the committee size (due to the reconstruction guarantees of VSS). We find that this increase is often small even for substantial drop out rates. For example, to tolerate a $10\%$ drop rate of honest committee members we need only increase the committee size from $46$ to $60$ (though this number depends on the algorithm, see Appendix B.1.1 for detailed analysis).

**Subsampling Clients.** This setting inherits the security of our original protocol as long as all honest peers agree on the selected subsample of clients in each round. This can be accomplished efficiently via secure coin flipping [8], e.g., before the protocol commences. Then, our protocol's computation is reduced proportionally to that of execution on the subsample.

# 5  Lifting Robust-Aggregation Algorithms to a Malicious-Security Model

Having discussed how a single-server robust aggregation with a computationally surjective $F^P$ can be lifted to the malicious P2P setting with high efficiency, we apply this principle to the design of malicious-secure versions of three popular robust aggregation algorithms: robust stochastic aggregation (RSA) [32], centered clipping (CC) [29], and FLTrust (FLT) [14] in the P2P setting.

## 5.1 Instantiating Robust Stochastic Aggregation (RSA) in our malicious-secure framework.

RSA is a lightweight algorithm for Byzantine-robust convex optimization [32] (see Appendix B.4.1 for a summary). We observe that it can be lifted to the malicious security model with high efficiency with very few modifications to the algorithm because it is computationally surjective (which we show formally in Appendix B) and the underlying MPC can be efficiently instantiated.

In RSA peer updates are the sign of the difference between each parameter of the local and global models. In other words, the $F^P$ of RSA gives $V = \{-1, 1\}^d$, where $d$ is the number of parameters in the model. Thus, it is sufficient for peers to prove in zero-knowledge that their updates are in the set $V = \{-1, 1\}^d$. This can be accomplished efficiently by having each peer represent their update as $d$ shares of binary values. The committee can perform a distributed zero knowledge (DZK) proof that a shared $x$ is binary-valued by constructing shares of $x \cdot (1 - x)$ and revealing it to be zero. These proofs can be batched together for a substantial improvement in efficiency. In particular, for every shared value $x_i$, parties uniformly sample a random value $r_i$, and locally construct shares of the sum $\sum r_i \cdot (x_i \cdot (1 - x_i))$. The parties then reconstruct the sum—if it is 0, then each of the $(x_i \cdot (1 - x_i))$ components must have been 0 with all but negligible probability. For a more detailed treatment of this technique, see [12].

During the computation of $F^R$, the committee needs only to sum the shares and send out the reconstructed sum. The actual value of the summed updates in $\{-1, 1\}$ is implicitly given by the sum of the binary values (if the sum of the binary values is $x$, simply take $2x - m$).

## 5.2 Instantiating Centered Clipping (CC) in our malicious-secure framework.

CC with momentum is a robust aggregation algorithm that ensures protection against time-coupled poisoning attacks [29] (see Appendix B for a summary). To lift it to our improved security model with

practical efficiency, we construct a computationally surjective variant of the CC algorithm. Namely, while canonical CC clips local updates using the $\ell_2$ norm, we use the $\ell_\infty$ norm.[4] In other words, we clip the gradients to a $\tau$-*box* rather than a $\tau$-*ball*. This modification admits a computationally surjective $F^P$ with an efficient DZK proof that a client update is within the valid domain. In particular, we take $V = [0, 2^\theta - 1]^d$. Then in $F^P$ we scale, round, and map clipped gradient updates to be within this domain. Here $\theta$ is a public constant large enough to limit discretization error of local updates during scaling—in experiments with CC we set $\theta$ to 32 in order to align with 32-bit fixed-point numbers. Smaller values of $\theta$ will increase protocol efficiency, at the expense of higher discretization error during rounding and mapping in $F^P$ step 3. The computational surjectivity of this $F^P$ follows from a similar argument to Lemma 2 (see Appendix B).

**DZK Proof of Valid Update.** We specify that local updates $v_i$ are submitted as vectors of the individual component bits of the processed gradient update. This means that each bit will be individually secret shared, which allows the committee to verify whether each one is binary-valued (using the same DZK technique described above for the RSA protocol). Since we scaled each update to fit within a $2^\theta$-sized $d$-dimensional box, the $d$ sets of $\theta$ binary values in the update trivially encode a point within the box. Thus, a proof that each component of the bitwise update is binary-valued equates to a proof that the update is in $V$.

The global update is aggregated by summing the bits at each position of the client update vectors. The sums are reconstructed and sent directly to all clients. They implicitly encode the updated global parameters $w'$, which are recovered via client-side computation in order to keep the computation of $F^R$ light-weight. Details of our malicious-secure Centered Box Clipping protocol can be found in Figure 8 (in Appendix).

## 5.3 Instantiating FLTrust in Malicious-Secure Framework

FLTrust (FLT) is a robust aggregation algorithm that uses a trusted dataset to filter out poisoned updates [14] (see Appendix B for a summary). As with CC, we construct a tailored variant of FLT that admits a computationally surjective $F^P$ to improve efficiency. In particular we rotate and scale the "root" update $g_0$ to be a unit vector aligned with the x-axis. This allows us to take $V$ to be the set of unit vectors in the half-space defined by a non-negative x-coordinate. As such, $F^P$ involves scaling and rotating client updates so that the angle between them and $g_0$ is preserved. Similarly to CC, we encode client updates as $\theta$-bit fixed point numbers. In our benchmarks for FLT, we set $\theta$ to 16 to compensate for the increased memory demands of this protocol. We use a committee size of 121 in order to enable multiplication of secret shared values (see Appendix B for details).

**DZK Proof of Valid Update.** As in CC, the magnitudes of local updates $v_i$ are submitted as shares of each bit in the binary representation of each fixed-point number. Clients additionally submit shares encoding sign for each parameter, with the exception of the x-coordinate, which is assumed to be always non-negative. We use the previously described technique to verify that the shares encoding magnitude are binary-valued. We use a similar technique to verify that shares encoding sign are in the set $\{-1, 1\}$ (i.e. we reveal $(b+1)(b-1)$ to be zero using a batch check). Further, we verify that submitted updates are unit length by constructing shares of $\langle \bar{g}_i, \bar{g}_i \rangle - C$, where $C$ is the squared length of a unit vector represented as a $\theta$-bit fixed-point number. Revealing this quantity to be zero verifies in zero-knowledge that $\bar{g}_i$ was indeed unit length.

# 6 Verifying Empirical Efficacy and Efficiency

Our empirical evaluation focuses on exploring three major axes: (1) the Byzantine robustness of our implementations due to modifications we introduced, (2) the computational efficiency of our protocol, and (3) the tradeoff between computational efficiency and Byzantine robustness. To this end, we center our comparisons on robust stochastic aggregation (RSA), Centered Clipping (CC), and FLTrust (FLT) but remark that our framework is compatible with other (potentially future) Byzantine robust algorithms as well. We demonstrate the practical efficiency of our case studies in the P2P Learning framework while maintaining the same robustness of the algorithms as in their clear versions.

---

[4] Karimireddy et al. [29] proved CC is robust under clipping for the $\ell_p$ norm for arbitrary choice of real numbers $p \geq 1$, which do not extend to the $\ell_\infty$ norm. We show empirically that centered clipping with the $\ell_\infty$ norm achieves similar model fidelity against known attacks in Appendix B.

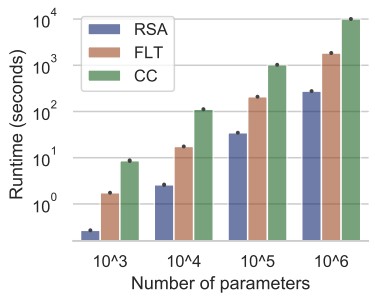

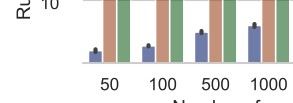

(a) Runtime vs Number of Parameters.
(b) Runtime vs Number of Peers.

Figure 5: **Computational Efficiency vs Number of Parameters and Peers.** We report CPU wall-clock time for the execution of the aggregation step of our protocol – the computation of $F^R$ in a single training round. The runtime performance of the algorithms (RSA, FLT, and CC) scales linearly with the number of parameters and peers. When modifying parameters we use a total of $100$ peers (left subfigure) and $10^5$ parameters set when changing the number of peers (right subfigure). For RSA and CC, the aggregation committee size is set to $46$, and for FLT it is set to $121$ in order to accommodate the secret share multiplications of the protocol (see Appendix B for details).

## 6.1  Security Does not Impact Robustness

We verify if the properties of the robust aggregation algorithms hold after the required modifications to lift them to the malicious setting, e.g., switching to fixed point numerical precision. In Figure 4, we use the IID MNIST dataset and 20 peers, of which there are 10 malicious workers[5]. We compare the robustness of CC against the ALIE (A Little Is Enough) attack [2] before and after lowering CC's numerical precision. We observe that the algorithm preserves its robustness despite the required changes. We also present corresponding additional studies (e.g. comparison between $\ell_2$ and $\ell_\infty$ norm for CC) in Appendix B. We observe that all the modified algorithms, namely CC, FLT, and RSA exhibit comparable performance to the original algorithms.

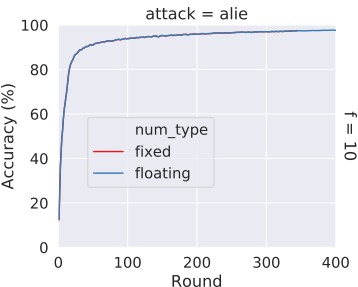

Figure 4: **Fixed vs floating-point numerical precision for CC.**

## 6.2  Scaling of Computational Efficiency

Because P2P learning algorithms typically require upwards of $1000$ rounds of the protocol to converge, it is a necessity to have an efficient protocol. In Figure 5, we analyze the two major factors influencing this: the size of the vector (ML model) being aggregated (denoted as the number of parameters), and the number of peers participating in the collaborative learning. We observe much better performance for RSA than other algorithms per training round. This results from a more concise form of the information exchanged between peers in the case of RSA, where local updates from each peer are represented as an array of bits. In contrast, model updates sent between peers in FLT or CC are always encoded as fixed points, 16 for FLT vs 32 for CC. The more efficient encoding in RSA provides a speedup of around $\sim$30X in comparison to CC and $\sim$6X over FLT. Our framework scales efficiently to even 5000 participants; we observe a linear growth in terms of the elapsed time per training round. Similarly, the computation time scales linearly for RSA, FLT, and CC, with the number of parameters. We further compare the communication cost between frameworks in Appendix B.

## 6.3  End-to-end Protocol Evaluation in Presence of Attacks

We estimate the accuracy and runtime of the modified algorithms in the presence of different types of attacks in Figure 6. We compute the number of rounds to convergence, and use the per-round CPU

---

[5]Note, this is a higher adversarial proportion than can be tolerated by our end-to-end framework due to the crytographic elements of our framework. We include this evaluation because the Byzantine robustness literature commonly considers this regime. This evaluation ensures that our underlying aggregation algorithms meet these standards (even when modified for efficiency). Thus, we also benchmark *accuracy* and *robustness* of the aggregation algorithms outside of the cryptographic elements, finding that the robustness guarantees are retained.

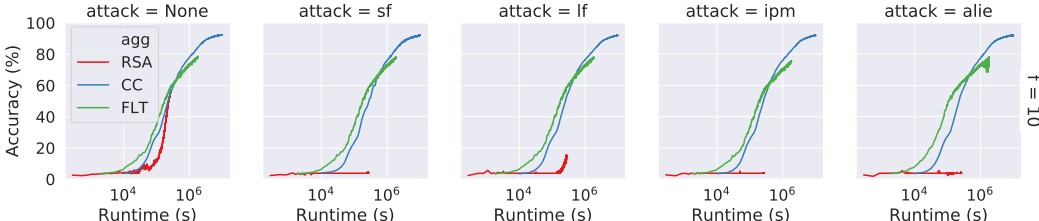

Figure 6: **Byzantine Robustness of P2P Learning Protocols** for iid EMNIST. We compare RSA, FLT, and CC after their instantiations in our framework. A cohort size of 50 peers is used, of which there are 10 malicious workers. We consider four attacks and have a baseline without any malicious workers. We run each algorithm until its completion. CC achieves the highest final accuracy. FLT and CC converge much faster than RSA.

time for computation of $F^R$ in each algorithm, to estimate overall training runtime and accuracy for EMNIST (and similar results for MNIST in Appendix B). We plot the test accuracy (%) on the y-axis and the x-axis represents the estimated CPU time (measured in seconds, note that this is in the logarithmic scale) of the P2P training. We observe that in all cases, CC and FLT algorithms outperform RSA in terms of convergence speed and achieve higher final accuracy. Note that the overall convergence speed is decided by both the number of iterations of training and the cost of each iteration. Although RSA is faster to compute for one iteration due to reduced information exchanged in each iteration, it requires much more iterations than CC and FLT, and hence slower to converge. When considering only utility, CC also outperforms FLT consistently; however, under computation constraints, it is often the case that FLT is more efficient than CC. This is primarily because we use a fixed-point length ($\theta$) of 16 bits in the experiments for FLT, but 32 bits for CC.

## 7 Limitations

We provided a reference implementation of our protocol for three popular robust aggregation algorithms, namely RSA, FTL, and CC. We hope that our framework will be easy to extend to future robust aggregation methods. We acknowledge that operating in the malicious threat model also increases the cost of computation, communication, and storage, in comparison to the fully trusted environment or an honest-but-curious threat model.

Our protocol is focused on confidentiality and security of the training protocol when combined with robustness. This is one component of privacy-preserving machine learning that is critical to preventing many attacks (as outlined in Table 1 and Section 2). However, this does not prevent the privacy leakage obtained via interactions with the final trained model. For this, differential privacy (DP) [21], in particular DP machine learning techniques [1, 3, 18, 19] are required. Incorporating these techniques within our framework is of interesting future work.

## 8 Conclusions

The benefits of collaborative learning make it an attractive new paradigm that is increasingly adopted in many domains, such as the financial sector to enable collaboration between banks. However, there are many risks associated with collaboration due to clients or server(s) being actively malicious. Malicious clients can submit corrupted updates which leads to the failure of creating a useful shared model. Conversely, the leakage of the client's local data when contributing model updates has been demonstrated to be particularly strong when a central party cannot be trusted to orchestrate the collaborative learning protocol. To mitigate these issues, we propose a Peer-to-Peer Learning protocol that is robust against malicious clients and server(s) to train a shared model *without* a central party. We prove the cryptographic security of our protocol, providing the necessary security guarantees. Our novel framework is designed as a generic compiler that can efficiently convert robust aggregation algorithms to the P2P learning setting with the guaranteed malicious-secure protocol. We show empirically that the generated protocols retain their robustness guarantees. This generic approach can be applied to many (possibly future) aggregation algorithms.

## Acknowledgement

We would like to acknowledge our sponsors, who support our research with financial and in-kind contributions: Amazon, Apple, CIFAR through the Canada CIFAR AI Chair, DARPA through the GARD project, Intel, Meta, NSERC through the Discovery Grant, the Ontario Early Researcher Award, and the Sloan Foundation. Resources used in preparing this research were provided, in part, by the Province of Ontario, the Government of Canada through CIFAR, and companies sponsoring the Vector Institute. Xiao Wang is supported by NSF #2016240, #2236819, #2318974, and research awards from Google and Meta. Jha is supported by Air Force Grant FA9550-18-1-0166, the National Science Foundation (NSF) Grants CCF-FMitF-1836978, IIS-2008559, SaTC-Frontiers-1804648, CCF-2046710, CCF-1652140, and 2039445, and ARO grant number W911NF-17-1-0405, and DARPA-GARD problem under agreement number 885000. Franzese is supported by the National Science Foundation Graduate Research Fellowship Grant No. DGE-1842165. We would also like to thank CleverHans lab group members for their feedback.

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

# A  Broader Impacts

The goal of our work is to provide a protocol that enables collaborative learning with guaranteed confidentiality of client data and fidelity of the trained model, even when both clients and server(s) can act maliciously. A potential positive impact of this work is increased privacy and accountability in machine learning systems. One potentially negative impact could be the degradation of performance (in terms of compute time, communication overhead, or additional storage) for legitimate users. However, as shown in our experimental results, we are still able to cater to 100s of users with a model size of 1 mln parameters.

# B  Additional Information

We further present additional information, experimental results, as well as a comparison between RSA, Centered Clipping with Momentum, and FL Trust.

## B.1  Committee Size

The main protocol proceeds by first selecting a subset from the pool of peers which will be responsible for aggregating the updates of all the peers. This subset is termed the *aggregation committee*. To guarantee security, the size of the committee $m$ has to be adjusted based on the number of corrupted parties. Let us denote the set of corrupted parties as $\mathcal{B}$ with $|\mathcal{B}| = b$. If the committee members are selected randomly, then with probability $p = b/n$, a given committee member is an adversary. To ensure security in the malicious case, we need the aggregation committee to have an honest majority except with negligible probability (i.e. occurring with probability less than $2^{-40}$ as in [33, 42]). We can assess the probability of this event by modeling the number of corrupted peers in a uniform sample as a binomial random variable $X$ with bias $p = \frac{b}{n}$ and $m$ trials. In particular, we are interested in values of $p$ and $m$ for which $Pr[X \geq n/2] < 2^{-40}$. These values can be computed from the cumulative density function of the binomial distribution. Assuming a 10% adversarial corruption threshold (i.e. setting $p = 1/10$), we obtain a committee size of 46. We use this committee size for experiments with RSA and CC. With FLTrust, in order to accommodate secret share multiplications with Shamir secret sharing, we guarantee $Pr[X \geq n/3] < 2^{-40}$, which gives a committee size of 121.

### B.1.1  Tolerance of Committee Members Dropping

In general, our protocol requires that the number of adversaries in the aggregation committee be kept below a certain proportion in order to guarantee security. The committee size is chosen as the smallest number of parties such that (except with negligible probability) a random sample from the pool of clients has less than $1/2$ adversarial proportion (in the case of RSA, CC), or less than $1/3$ (in the case of FLT). To tolerate drop out of honest committee members, we simply need to select an increased committee size such that the proportion of adversaries in the committee stays beneath these thresholds even if some number of the honest parties drop out. In particular, if we choose a committee size which guarantees (except with negligible probability) that a random sample from the pool of clients has less than $\frac{1}{2} - \frac{q}{2}$) adversarial proportion, where $q$ is the proportion of tolerated dropouts from honest parties, we will guarantee that the adversarial proportion with reference to the number of committee members that stay online is at most $1/2$. We can find the necessary committee sizes by reasoning with the binomial distribution similarly to our original analysis of committee size. For example, to tolerate 5%, 10%, and 15% dropout of honest committee members, RSA and CC would require committee sizes of 53, 60, and 69 respectively (compared to 46 with no dropout tolerance), and FLT would require 157, 218, and 326 respectively (compared to 121 with no dropout tolerance).

## B.2  Building Blocks

**Byzantine Robust Aggregation.** In collaborative learning (e.g., federated learning), many clients submit model updates based on their local data. These local updates are aggregated to update the global model. In settings where clients are untrusted, some *Byzantine* or malicious clients may submit poisonous updates (which may take *arbitrary* values) with the aim of degrading the quality of the global model. Broadly speaking, Byzantine robust aggregation algorithms (often abbreviated to "robust aggregation"), guarantee that an accurate global model is trained as long as the proportion of

malicious clients is bounded by a certain threshold (e.g., the theoretical analysis in CC assumes a maximum of 15% malicious clients). Further formalization of this idea occurs in a variety of ways across different works of literature – our framework is intentionally modular, inheriting the guarantees of a given underlying algorithm. Our main contribution is augmenting robustness to malicious clients with the additional guarantee that aggregation is computed *correctly and confidentially* even in the presence of malicious servers / aggregation committee members. We use the cryptographic primitives reviewed below to achieve this guarantee.

**Committee Election.** Uniform election of committee members can be efficiently instantiated using coin-flipping [8]. A classical way to accomplish this is to have all peers generate a string of random bits locally. The peers then make a cryptographic commitment to their random bits and distribute it to all other peers. After all peers have made their commitments, the random bits are all revealed. The concatenation of all the random bits can then be used as input to a random oracle, whose outputs can be used to select the committee members uniformly at random. This method for uniform random committee election is secure as long as at least one peer behaves honestly during the commitment process. We leverage this to guarantee that the aggregation committee has an honest majority (or is 2/3 honest in the case of FLT) (see Appendix B.1 for details).

**Verifiable Secret Sharing.** To make our protocols secure in the presence of malicious adversaries, we require Verifiable Secret Sharing (VSS). A VSS scheme allows the secret owner with a secret $s$, to distribute shares of $s$ among $n$ parties with a threshold $t$ such that (a) any group of $t$ parties can reveal no information about $s$ and (b) any $t + 1$ parties can recover the correct value of $s$. In this work, we use Shamir secret sharing to instantiate VSS. Secrets are shared among members of the aggregation committee $C$. We make guarantees on the adversarial composition of $C$, and set $t$ such that honest parties may perform computations necessary during DZKP and MPC protocols (see below), yet adversarial parties never gain access to enough shares to reveal or modify $s$.

**Distributed Zero Knowledge Proofs.** A malicious-secure zero knowledge proof protocol enables a prover in possession of a witness $w$ to prove to a verifier that for some publicly known function $f$, $f(w)$ takes a particular value. It is guaranteed that the verifier learns no additional information about $w$ other than what is implicitly revealed by $f(w)$, and that no malicious prover can convince the verifier that $f(w)$ takes an incorrect value. A *distributed* zero-knowledge proof (DZKP) is a variation on this primitive, wherein the prover distributes secret shares of $w$ among a set of verifiers. Leveraging the linear operations on secret shares enabled by this setting can admit particularly efficient zero-knowledge proofs (see e.g. [12]). In our implementations, we use DZKP protocols which assume that the set of verifiers has an honest majority.

**Secure Multiparty Computation.** A malicious-secure multiparty computation (MPC) protocol enables a group of parties $P_1, P_2, \ldots, P_n$, with respective private inputs $x_1, x_2, \ldots, x_n$ to securely compute a function $f$ and obtain output $f(x_1, x_2, \ldots, x_n)$. In particular, it is guaranteed that no party learns any additional information about the inputs beyond what is implicit in the output, and it is guaranteed that $f$ is computed correctly, even in the presence of parties that behave in arbitrarily malicious ways. In our implementations, we use MPC protocols which assume that the set of parties has an honest majority (2/3 honest in the case of FLT).

**Composition.** All of the building blocks listed above are secure under universal composability [13] and thus their compositions (i.e., using them together, either in sequence or in parallel) are secure. They can all be implemented under information theoretic security, although we used a pseudorandom generator to minimize the communication. That is, using these primitives together in concert in our protocol preserves the security guarantees afforded by the individual building blocks.

To provide a concrete sense of how the security guarantees and trust assumptions of the building blocks work together in the full protocol, we provide a step-by-step elaboration on the protocol below.

1. **Committee Election** All clients use the method described above to randomly select the aggregation committee $C$ with malicious security. The analysis in Appendix B.1 guarantees that $C$ has honest majority.
2. **Client Local Computation.** Each client computes $F^C$ and $F^P$ to obtain a preprocessed local model update given their data, and the global model parameters.
3. **Verifiable Secret Sharing of Updates.** Each client secret shares their update with threshold $|C|/2$ ($|C|/3$ for FLT), and sends a share to each committee member. Since $C$ has honest majority, this guarantees that adversaries cannot alter or reveal the client updates.

4. **DZKP of valid update.** Clients prove to the committee that their updates are valid using a DZKP protocol that takes the secret shares as input. E.g. in P2P RSA, client updates must be binary-valued, so committee members create shares of a check value which is guaranteed to be $0$ if the update was binary-valued, while leaking no further information (see Appendix B.4.1 for details). Here security follows from the security of the DZKP and the VSS schemes.

5. **MPC for computing global updates.** Committee members compute $F^R$ in MPC, using client shares as input, to obtain a global model update. E.g. in P2P RSA, committee members sum the shares of all client updates using the standard secure addition protocol on Shamir secret shares. The committee members then reconstruct the shared sum to obtain the global update (correct reconstruction is guaranteed by VSS). Security follows from the security of the MPC and VSS schemes.

6. **Global updates sent to clients.** All committee members send the recovered value to all clients. Since $C$ has an honest majority, the clients are guaranteed to recover the correct global update by accepting the majority result.

## B.3 Security Proof

We provide a proof of Theorem 1 (malicious security of Figure 3) below.

*Proof:* We prove the security of the protocol by constructing a simulator interacting with the adversaries controlling a subset of the parties.

1 The simulator plays the role of coin flipping and return a uniform aggregation committee. If the committee contains more adversary than the allowed threshold, the simulator aborts.

The probability of simulator aborts in this step is negligible given the committee size and threshold.

2-4 The simulator obtains shares of $v_i$ from the adversary and sends them random shares on behalf of the honest parties.

5 If $F^P$ is not computationally surjective, The simulator plays the role of DZK to obtain the adversary's input $u_i$. If $F^P$ is computationally surjective, the simulator use $v_i$ to compute some $u_i$.

The simulator's running time is always polynomial in this step either because efficient extraction from DZK or because of the definition of computational surjectivity.

6 The simulator plays the role of DZK and check if $v_i$ is in the image of $F^P$ and aborts if it is not the case.

7 The simulator sends $u_i$ to $\mathcal{F}_{\text{P2PL}}$ and gets back the new updates; it then plays the role of $\mathcal{F}_{\text{MPC}}$ and sends back the new updates to the adversary.

$\square$

## B.4 Instantiating Our Malicious Framework

### B.4.1 Malicious-Secure P2P RSA.

**Overview of Single-Server RSA.** Single-server Byzantine-robust stochastic aggregation (RSA) [32] is a set of subgradient based algorithms for robust aggregation. The key component of the method is a regularization term incorporated into the objective function to make learning robust. To enable graceful handling of heterogeneous worker datasets, each client $i$ maintains a local set of model parameters $x_i^k$ whilst working together to optimize the global model parameters $w^k$ at a step $k$. At each step, clients compute a parameter update which takes into account their local data, their prior local model, as well as the global model parameters. The server receives the local client updates and uses the regularized objective to obtain a robust aggregate update. Client and server updates, respectively, are given by the equations:

$$x_i^{k+1} = x_i^k - \eta^k \left( \nabla F(x_i^k, \xi_i^k) + \lambda \text{sign}(x_i^k - w^k) \right) \tag{1}$$

$$w^{k+1} = w^k - \eta^k \left( \nabla f_0(w^k) + \lambda \left( \sum_{i \in [n]} \text{sign}(w^k - x_i^k) \right) \right) \tag{2}$$

where $\eta$ is a decaying learning rate hyper parameter, $\xi$ is a sampling of the local client dataset, $F(\cdot, \cdot)$ is the loss function, $f(\cdot, \cdot)$ is the robust ($\ell_2$) regularization term, $\lambda$ is a hyper parameter controlling the weighting of the robustness term, the $sign$ is performed element-wise, and $[n]$ is the set of clients.

**Lifting RSA to the P2P setting.** To cast RSA into our framework, we first observe that $\sum_{i \in [n]} \text{sign}(\boldsymbol{w}^k - \boldsymbol{x}_i^k)$ is the only term of the server's update that requires input from the clients. Thus we limit the work of the committee solely to computing this term, and the rest of the work is done locally. We instantiate RSA for our framework in Figure 7.

In the $F^C$ (client update computation) part of the RSA protocol, each peer receives the global model parameters $\boldsymbol{w}^k$. It computes local parameter update $\boldsymbol{x}_i^{k+1}$ based on the global model, the local model $\boldsymbol{x}_i^k$, and the local gradient $\nabla F$. In the $F^P$ (update preprocessing) part of the protocol, peers compute the sign of the difference between their local parameters and the global model parameters $\text{sign}(\boldsymbol{w}^k - \boldsymbol{u}_i)$, resulting in a bit vector $\boldsymbol{v}_i$ (one bit per model parameter). In the $F^R$ (aggregation) part of the protocol, the committee members receive secret shares of $\text{sign}(\boldsymbol{w}^k - \boldsymbol{x}_i^k)$ from each participant. We observe that RSA can be lifted to the malicious security model with high efficiency: it is provably computational surjective and the underlying MPC can be efficiently instantiated.

**Computational Surjectivity.** Recall that in RSA peer updates are the sign of the difference between each parameter of the local and global models (Figure 7). In other words, the $F^P$ of RSA gives $V = \{-1, 1\}^d$, where $d$ is the number of parameters in the model. In the single-server model of RSA [32] and in Figure 7, poisonous peers can choose arbitrary $\boldsymbol{u}_i$ before $F^P$ is computed, which gives $\boldsymbol{v}_i = \text{sign}(\boldsymbol{w}^k - \boldsymbol{u}_i)$. Now we are ready to show the computational surjectivity of this $F^P$.

**Lemma 1** $F^P$ described in Figure 7 is a computationally surjective function.

*Proof:* Fix an arbitrary point $\boldsymbol{v} = (v_1, \cdots, v_d) \in V = \{-1, 1\}^d$. We can construct $\boldsymbol{u} \in U$ that $F^P$ maps to $\boldsymbol{v}$ by first fixing some arbitrary $\boldsymbol{w}^k = (w_1, \cdots, w_d)$, and letting $\boldsymbol{u} = (u_1, \cdots, u_d)$ such that

$$u_j = w_j - v_j \text{ for each } j \in [d].$$

Clearly the $F^P$ of RSA $\boldsymbol{v}_i = \text{sign}(\boldsymbol{w}^k - \boldsymbol{u}_i)$ maps $\boldsymbol{u}$ to the arbitrary $\boldsymbol{v}$. So $F^P$ is computationally surjective. □

**Details of the cryptographic protocol.** Thus, following Figure 3, it is sufficient for peers to prove in zero knowledge that their updates are in the set $V = \{-1, 1\}^d$. This can be accomplished efficiently by having each peer represent their update as $d$ shares of binary values.

The committee can verify that a shared $x$ is binary-valued by constructing shares of $x \cdot (1 - x)$ and revealing it to be zero. We implement this step efficiently by batching the binary-value DZK proofs together. That is, for every shared value $x_i$, parties uniformly sample a random value $r_i$, and locally construct shares of the sum $\sum r_i \cdot (x_i \cdot (1 - x_i))$. The parties then reconstruct the sum – if it is 0, then each of the $(x_i \cdot (1 - x_i))$ components must have been 0 with all but negligible probability. For a more detailed treatment of this technique, see [12].

During the computation of $F^R$, the committee needs only to sum the shares and send out the reconstructed sum. The actual value of the summed updates in $\{-1, 1\}$ is implicitly given by the sum of the binary values (if the sum of the binary values is $x$, simply take $2x - m$). The updated global model parameters can then be obtained via local computation of Equation 2.

**Computational Surjectivity.** In RSA, peer updates are the sign of the difference between each parameter of the local and global models (Figure 7). The $F^P$ of RSA gives $V = \{-1, 1\}^d$, where $d$ is the number of parameters in the model. In the single-server model of RSA [32] and in Figure 7, poisonous peers can choose arbitrary $\boldsymbol{u}_i$ before $F^P$ is computed, which gives $\boldsymbol{v}_i = \text{sign}(\boldsymbol{w}^k - \boldsymbol{u}_i)$. Now we are ready to show the computational surjectivity of this $F^P$.

**Lemma 2** $F^P$ described in Figure 7 is a computationally surjective function.

*Proof:* Fix an arbitrary point $\boldsymbol{v} = (v_1, \cdots, v_d) \in V = \{-1, 1\}^d$. We can construct $\boldsymbol{u} \in U$ that $F^P$ maps to $\boldsymbol{v}$ by first fixing some arbitrary $\boldsymbol{w}^k = (w_1, \cdots, w_d)$, and letting $\boldsymbol{u} = (u_1, \cdots, u_d)$ such that

$$u_j = w_j - v_j \text{ for each } j \in [d].$$

Clearly the $F^P$ of RSA $\boldsymbol{v}_i = \text{sign}(\boldsymbol{w}^k - \boldsymbol{u}_i)$ maps $\boldsymbol{u}$ to the arbitrary $\boldsymbol{v}$. So $F^P$ is computationally surjective. □

$$\boldsymbol{u}_i \leftarrow F^C(\mathsf{data}, \mathsf{st}, \boldsymbol{w})$$

    1. $(k, \boldsymbol{x}_i^k) \leftarrow \mathsf{st}$

    2. Sample $\xi_i^k$ from local data $\mathcal{D}_i$

    3. $\boldsymbol{x}_i^{k+1} = \boldsymbol{x}_i^k - \eta^k \left( \nabla F(\boldsymbol{x}_i^k, \xi_i^k) + \lambda \mathrm{sign}(\boldsymbol{x}_i^k - \boldsymbol{w}^k) \right)$

    4. $\boldsymbol{u}_i = \boldsymbol{x}_i^k$

    5. $\mathsf{st} \leftarrow (k+1, \boldsymbol{x}_i^{k+1})$

$$\boldsymbol{v}_i \leftarrow F^P(\boldsymbol{u}_i)$$

    1. Set $\boldsymbol{v}_i = \mathrm{sign}(\boldsymbol{w}^k - \boldsymbol{u}_i)$

$$\boldsymbol{w} \leftarrow F^R(\{\boldsymbol{v}_i\}_{i \in [n]})$$

    1. Set $\boldsymbol{w}^{k+1} = \boldsymbol{w}^k - \eta^k \left( \nabla f_0(\boldsymbol{w}^k) + \lambda(\sum_{i \in [n]} \boldsymbol{v}_i) \right)$

Figure 7: **P2P Learning with RSA.** $F^R$ can be computed efficiently by performing only $\sum_{i \in [n]} \boldsymbol{v}_i$ on the committee side. The rest of the terms are public, so the remainder of the update can be computed locally.

### B.4.2 Malicious Secure P2P CC

**Overview of Single-Server Centered Clipping.** Centered Clipping [29] is a recent robust aggregation that ensures a high level robustness even when the noise distribution is not uni-modal (which is assumed in many prior works.) It also provides better robustness when corrupted updates at different rounds are correlated. Below we first discuss details of the algorithm and then how to express it in our framework.

**Centered Clipping (no momentum):** Given the training iteration $k$, globally shared model parameters $w^k$, local model parameters $x_i^{k+1}$ in client $i$, and a radius $\tau$, CC using the $\ell_2$-norm computes an updated weight vector as follows:

$$x_i^{k+1} = (x_i^{k+1} - w^k) \min\left(1, \frac{\tau}{||x_i^{k+1} - w^k||_2}\right) \tag{3}$$

$$w^{k+1} = w^k + \frac{1}{n} \sum_{i=1}^{n} x_i^{k+1} \tag{4}$$

In Equation (3), we clip the parameters for each client $i$, and then aggregate them in Equation (4).

**Centered Clipping with Momentum:** In addition to the above, each non-Byzantine client $i$ first computes a gradient update $\nabla F$ based on their mini-batch $\xi_i^k$ and the current global weights $w^k$. Then, using the momentum parameter $\beta$, each client computes a momentum vector as shown in Equation 5 (executed before Equation (3) and Equation (4)):

$$x_i^{k+1} = (1 - \beta)\nabla F(w^k, \xi_i^k) + \beta x_i^k \tag{5}$$

**Lifting CC to the P2P setting.** We bring CC into the P2P setting by placing the momentum computation inside $F^C$, the clipping operation inside $F^P$, and the aggregation of clipped updates in $F^R$. The clipping operation is performed on individual client updates, and thus can be performed on the client side. Further, as in RSA we note that $F^R$ is a linear function, and thus can be computed efficiently using the homomorphic addition and scalar multiplication properties of Shamir secret sharing.

Centered Clipping does not naturally give us a surjective $F^P$. Of note, if a corrupted peer supplies a value of $\boldsymbol{v}_i$ that is outside of the $\tau$-ball surrounding $\boldsymbol{w}$, the global update will be computed incorrectly and the model fidelity guarantees will be broken. To avoid this possibility, we make a slight modification to the CC algorithm. Namely, we clip local updates using the $\ell_\infty$ norm rather than the $\ell_2$ norm. In other words, we clip the gradients to a $\tau$-*box* rather than a $\tau$-*ball*. The computation of the global update thus becomes

$$\boldsymbol{w}_{k+1} = \boldsymbol{w}_k + \frac{1}{m}\sum_{i=1}^{m}\min(\tau, \max(-\tau, x_i - \boldsymbol{w}_k)) \tag{6}$$

This modification admits a computationally surjective $F^P$ with an efficient DZK proof that a client update is within the valid domain. In particular, we take $V = [0, 2^\theta - 1]^d$. Then in $F^P$ we scale, round, and map clipped gradient updates to be within this domain. Here $\theta$ is a public constant large enough to limit discretization error of local updates during scaling – in the present study we set $\theta$ to 32 in order to align with 32-bit fixed-point numbers. Smaller values of $\theta$ will increase protocol efficiency, at the expense of higher discretization error during rounding and mapping in $F^P$ step 3. Computational surjectivity of this $F^P$ follows from a similar argument to Lemma 2.

**DZK Proof of Valid Update.** We specify that local updates $\boldsymbol{v}_i$ are submitted as vectors of the individual component bits of the processed gradient update. This means that each bit will be individually secret shared, which allows the committee to verify whether each one is binary-valued (using the same DZK technique described above for the RSA protocol). Since we scaled each update to fit within a $2^\theta$-sized $d$-dimensional box, the $d$ sets of $\theta$ binary values in the update trivially encode a point within the box. Thus, a proof that each component of the bitwise update is binary-valued equates to a proof that the update is in $V$.

The global update is aggregated by summing the bits at each position of the client update vectors. The sums are reconstructed and sent directly to all clients. They implicitly encode the updated global parameters $\boldsymbol{w}'$, which are recovered via client-side computation in order to keep the computation of $F^R$ light-weight. Details of our malicious-secure Centered Box Clipping protocol can be found in Figure 8.

### B.4.3 Malicious Secure P2P FLTrust

**Overview of Single-Server FLTrust.** Single-server FLTrust (abbreviated FLT) [14] is a robust aggregation algorithm that bootstraps trust using a clean "root" dataset maintained by the server. During each iteration, the server compares client gradients against the gradient computed from the root dataset. Specifically, the server computes a 'trust score' (TS) for each client gradient $i \in [m]$, which it uses to compute a weighted sum of normalized gradients which makes up the final aggregate. The trust score and update aggregation are given by the following equations:

$$TS_i = ReLU\left(\frac{\langle g_i, g_0 \rangle}{||g_i||||g_0||}\right) \tag{7}$$

$$g = \frac{1}{\sum_{j=1}^{m} TS_i}\sum_{i=1}^{m} TS_i \cdot \bar{g}_i \tag{8}$$

$$w = w + \alpha \cdot g \tag{9}$$

Where $TS_i$ is the trust score for client $i$, $g_i$ is the local gradient for client $i$, $g_0$ is the gradient computed from the root dataset, and $\bar{g}_i$ is the gradient of client $i$ normalized to have the same length as $g_0$. As a brief explanation of the framework, the trust score acts as a clipped version of the cosine similarity – the greater the angle between $g_i$ and $g_0$, the smaller the scaling factor that weights $\bar{g}_i$ in the weighted sum. The ReLU ensures that any $g_i$ with a negative cosine similarity is clipped to 0, and thus contributes no weight to the sum.

**Lifting FLT to the P2P setting.** We begin by assuming that the root dataset $D_0$ is publicly accessible, so that all clients may compute the root update $g_0$ locally, in addition to their local update $g_i$ inside of $F^C$. In $F^P$ we perform normalization and rotation to simplify the computation of Equations 7 and 8 in $F^R$ (explained in more detail below). In $F^R$, we securely compute the trust score of each client and the corresponding weighted sum of gradients. This weighted sum is submitted as the global update – computation of the updated model parameters is left to the clients as a post-processing step.

The representation of $\boldsymbol{v}_i$ is chosen to enable efficient computation of $F^R$ and of DZK proofs of update validity. In detail, we perform a rotation of $g_i$ and $g_0$ such that $g_0$ is aligned with the $x$-axis (and the angle between $g_0$ and $g_i$ is preserved). We also normalize such that $g_0$ and $g_i$ are unit-length. Further, when submitting client updates we use a representation that can only encode a non-negative $x$-coordinate (by decomposing each entry of $g_i'$ into a sign and magnitude, and only

$\boldsymbol{u}_i \leftarrow F^C(\mathsf{data}, \mathsf{st}, \boldsymbol{w})$

1. $(k, \boldsymbol{m}_i^k) \leftarrow \mathsf{st}$
2. Parse $\boldsymbol{w} = \left\{b_j^{\boldsymbol{w}}\right\}_{j \in [d \cdot \theta]}$ into $d$ sets of $\theta$ values each, corresponding to the $d$ parameters of the model. Index them as $p_{ih}$ where $i \in [\theta]$ and $h \in [d]$. // Parse and index the bitwise global update to align with parameters of the model
3. For $h \in [d]$, $s_h \leftarrow \sum_{i \in [\theta]} p_{ih} \cdot 2^i$ // intermediate value of global update reconstruction
4. $w_h^k \leftarrow w_h^{k-1} - \eta(\frac{1}{m} \cdot s_h)$ for $h \in [d]$. Call $\boldsymbol{w}' \leftarrow \left\{w_h^k\right\}_{h \in [d]}$ // reconstruct global model parameters
5. Sample $\xi_i^k$ from local data $\mathcal{D}_i$
6. Compute $\boldsymbol{u}_i = (1 - \beta^k)(\nabla F(\boldsymbol{w}', \xi_i^k)) + \beta^k \boldsymbol{m}_i^k$
7. $\mathsf{st} \leftarrow (k+1, \boldsymbol{u}_i)$

$\boldsymbol{v}_i \leftarrow F^P(\boldsymbol{u}_i, \boldsymbol{w})$

1. Compute $\boldsymbol{v}_i''' \leftarrow \min(\tau, \max(-\tau, \boldsymbol{u}_i - \boldsymbol{w})) + \tau$
2. Compute $\boldsymbol{v}_i'' \leftarrow \frac{(\boldsymbol{v}_i'' - \boldsymbol{w})}{\tau} \cdot 2^{\theta-1}$ // scale the clipped value and center it to the origin
3. Round and map entries of $\boldsymbol{v}_i''$ to unsigned $\theta$-bit integers values $\in [0, 2^\theta - 1]$. Call the result $\boldsymbol{v}_i'$.
4. Decompose $\boldsymbol{v}_i'$ into the component bits used to represent each value in the vector, indexed as $b_{ij}$ for $j \in [d \cdot \theta]$. Submit a vector of the individual bits as $\boldsymbol{v}_i$.

$\boldsymbol{w} \leftarrow F^R(\{\boldsymbol{v}_i\}_{i \in [n]}, \boldsymbol{w})$

1. $(k) \leftarrow \mathsf{st}$
2. For $j \in [d \cdot \theta]$, compute $b_j^{\boldsymbol{w}} \leftarrow \sum_{i \in [m]} b_{ij}$. // sum each bit across client updates
3. $\boldsymbol{w} \leftarrow \left\{b_j^{\boldsymbol{w}}\right\}_{j \in [d \cdot \theta]}$
4. $\mathsf{st} \leftarrow (k+1)$

Figure 8: Centered Box Clipping. By clipping to a box and scaling that box to size $2^\theta$, this modification of Centered Clipping achieves computational surjectivity and an efficient proof to verify that shared peer updates are inside $V$.

accepting a magnitude – and not a sign bit – for the $x$-coordinate). This canonical representation simplifies computation of the trust score. In particular, since $g_0$ and $g_i$ are normalized to unit vectors, computation of the cosine similarity $\frac{\langle g_i, g_0 \rangle}{||g_i||||g_0||}$ simplifies to $\langle g_i, g_0 \rangle$, and since $g_0$ is aligned with the $x$-axis, this further simplifies to selecting the $x$-coordinate of $g_i$. Further, we avoid taking the ReLU within $F^R$ by choosing a representation of $\boldsymbol{v}_i$ that cannot represent a $g_i$ with negative $x$-coordinate, and specifying that any honest party whose local gradient has negative $x$-coordinate supplies an update that will have 0 weight during the computation of Equation 8 (we use the symbol $\perp$ as a placeholder for such an update – in practice, this can be any arbitrary unit vector with 0 in the $x$-coordinate). Thus computation of the trust score during $F^R$ is simplified to taking the $x$-coordinate of $g_i'$.

The chosen representation of $\boldsymbol{v}_i$ constrains the image of $F^P$ to the set of unit vectors with non-negative $x$-coordinates. If we restrict the codomain of $F^P$ to this set, we achieve computational surjectivity. This follows from a simple argument:

*Proof:* Fix an arbitrary point $\boldsymbol{v}$ in the set of unit vectors with non-negative $x$-coordinates. Fix an arbitrary $g_0$. Let $M$ be a rotation matrix that rotates $g_0$ to the $x$-axis. Consider a client update $\boldsymbol{u}$ such that $M\boldsymbol{u}$ is on the line extending from the origin to $\boldsymbol{v}$. By definition, $F^P$ maps $\boldsymbol{u}$ to $\boldsymbol{v}$. □

Finally, we construct DZK proofs to verify that $\boldsymbol{v}_i$ falls inside the set of unit vectors with non-negative $x$-coordinates.



**Inputs / Public Constants:**

• Assume all client states $S$ contain a public root dataset $D_0$ (in addition to their private dataset $D_i$)

---

$\boldsymbol{u}_i \leftarrow F^C(\mathsf{data}, \mathsf{st}, \boldsymbol{w})$

1. $g_0 \leftarrow \mathtt{ModelUpdate}(\boldsymbol{w}, D_0)$ // compute update from root dataset, save in client state
2. $g_i \leftarrow \mathtt{ModelUpdate}(\boldsymbol{w}, D_i)$ // compute local update from client dataset
3. $\boldsymbol{u}_i \leftarrow g_i$

---

$\boldsymbol{v}_i \leftarrow F^P(\boldsymbol{u}_i, \boldsymbol{w})$

1. $\bar{g}_0 \leftarrow \frac{g_0}{\|g_0\|}$ // normalize to unit length
2. $\bar{g}_i \leftarrow \frac{\boldsymbol{u}_i}{\|\boldsymbol{u}_i\|}$ // normalize to unit length
3. $M \leftarrow$ rotation matrix aligning $\bar{g}_0$ with the x-axis.
4. $\bar{g}_0 \leftarrow M\bar{g}_0$
5. $\bar{g}_i \leftarrow M\bar{g}_i$ // rotate client update by the same angle
6. Represent $\bar{g}_i$ and $\bar{g}_0$ as $\theta$-bit fixed-point numbers with a designated sign bit, call this representation $g_i'$ and $g_0'$.
7. If the $x$-coordinate of $\bar{g}_i'$ is negative, submit $\perp$ as $\boldsymbol{v}_i$.
8. Otherwise, submit a vector of the individual bits of $\bar{g}_i'$ as $\boldsymbol{v}_i$. *Each coordinate should be submitted as a sign bit and a binary-encoded magnitude, except for the x-coordinate which should only have a magnitude since it is non-negative.*

---

$\boldsymbol{w} \leftarrow F^R(\{\boldsymbol{v}_i\}_{i \in [n]}, \boldsymbol{w})$

1. Parse $\boldsymbol{v}_i$ appropriately as $\bar{g}_i'$
2. $TS_i \leftarrow$ magnitude of x-coordinate of $\bar{g}_i'$ // since $g_0$ is aligned with $x$-axis
3. Submit $\bar{g} \leftarrow \sum_{i \in [n]} \bar{g}_i' \cdot TS_i$ as global update. *Denormalization, rotation, and computation of global model parameters via $\boldsymbol{w} \leftarrow \boldsymbol{w} + \alpha \cdot g$ is performed as post-processing on the client side.*



Figure 9: FLTrust.

**DZK Proof of Valid Update.** As in RSA and CC, we perform a batch check that all submitted shares are binary-valued (see previous sections for details). We additionally perform a DZK proof that all updates are unit length, by constructing shares of $\langle \bar{g}_i', \bar{g}_i' \rangle - C$ and revealing them to be 0, where $C$ is a constant which encodes the square of a $\theta$-bit fixed point number with unit magnitude. We batch check these proofs by obtaining shared random field elements $r_i$ and constructing shares of the sum $\sum r_i \cdot (\langle \bar{g}_i', \bar{g}_i' \rangle - C)$, and finally revealing them to be 0 (i.e. using the same technique as described in the binary-value batch check for RSA). We also perform a DZK proof to ensure that the sign bits are in $\{-1, 1\}$ by computing shares of $(b - 1)(b + 1)$ and revealing them to be $0$ – this check is batched in the same way as the previous checks.

## B.5  Experimental Design

While lifting robust aggregation algorithms to the malicious-secure P2P Learning security model, we make small changes to the algorithms to tailor them for efficiency in the setting. Thus, in order to evaluate P2P Learning, we design experiments to test (1) the effectiveness (in terms of accuracy and robustness) of these tailored algorithms, as well as (2) the efficiency of their implementation as cryptographic protocols. These goals are performed using distinct code bases: we used PyTorch to benchmark accuracy and robustness, and we used the NTL package [40] in C++ to implement the local computation for the aggregation steps of our malicious-secure framework.

### B.5.1  Accuracy and Robustness Experiments

To benchmark the robustness of the different aggregation protocols evaluated in the paper, we ran experiments under each to train a central model in a collaborative machine learning setting with a

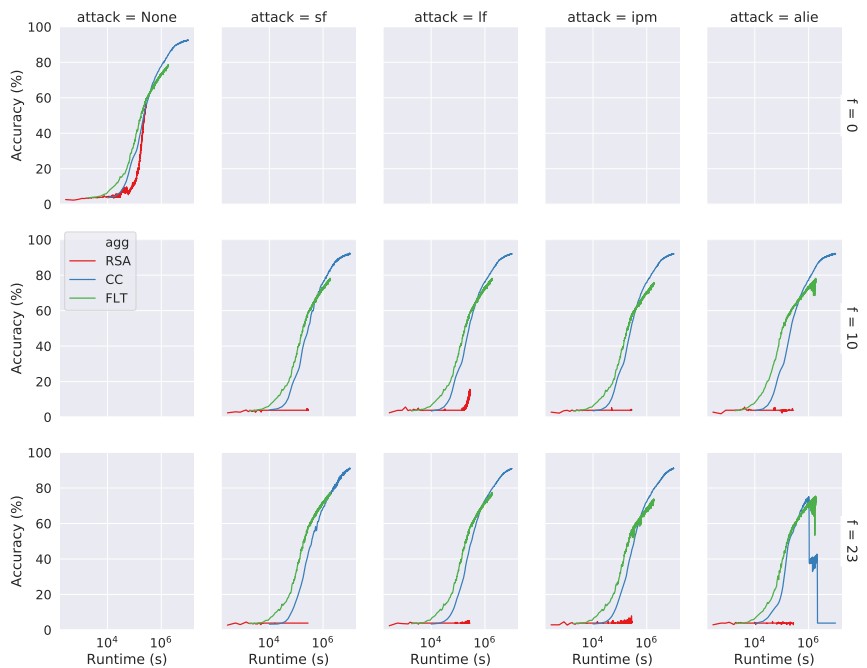

Figure 10: **Byzantine Robustness of Doubly Robust Protocols** for iid EMNIST. We compare RSA and CC after their instantiations in our framework. A cohort size of 50 peers is used. $f$ is the number of malicious workers. We run each algorithm until its completion.

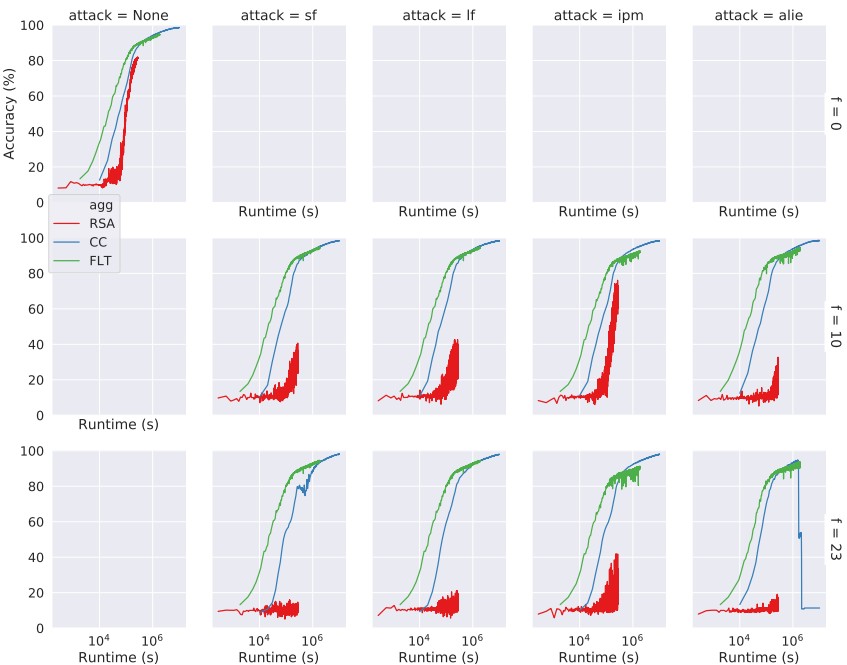

Figure 11: **Byzantine Robustness of Doubly Robust Protocols** for iid MNIST. We compare RSA and CC after their instantiations in our framework. A cohort size of 50 peers is used. $f$ is the number of malicious workers. We run each algorithm until its completion.

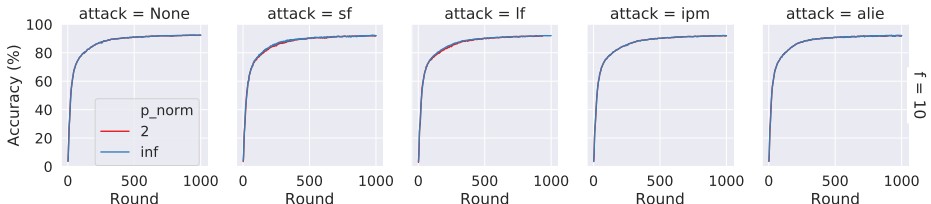

Figure 12: $\ell_2$ **vs** $\ell_\infty$ **norm for CC** for iid EMNIST.

| Dataset | Type | No attack | sf attack | lf attack | ipm attack | alie attack |
|---------|------|-----------|-----------|-----------|------------|-------------|
| **EMNIST** | **IID** | 91.68 | 91.09 | 90.83 | 91.20 | 91.43 |
| **EMNIST** | **nonIID** | 91.69 | 91.07 | 90.88 | 91.19 | 87.20 |
| **CIFAR100** | **IID** | 48.04 | 33.77 | 44.94 | 45.64 | 32.77 |

Table 2: Maximum accuracy achieved by CC in a given setting.

cohort size of 50 participants and varying numbers of malicious workers (0, 10, 23). 4 attacks, namely bit flip (bf) [45], label flip (lf) [6], inner product manipulation (ipm) [46], and "a little is enough" (alie) [2], were evaluated. In all cases, we computed the testing accuracy as a function of the number of rounds of training.

MNIST (Digits) and EMNIST (Letters) datasets were used as the datasets with the data being evenly divided among the peers. The model architecture from [30] (with 1.2M parameters) was used for MNIST and this architecture was modified to have 26 neurons in the last layer for EMNIST. During training, each client uses a local mini-batch of size 32 at each round and a learning-rate of 0.01.

The training experiments were repeated over two random seeds. The PyTorch [36] framework was used for all experiments.

### B.5.2 Computational Efficiency Experiments

To benchmark the efficiency of our framework, we wrote code to perform all local computation steps necessary to run the aggregation step for a single committee member ($F^R$) of malicious-secure P2P RSA, CC, and FLT. We used an `m5.metal` instance on Amazon EC2 to obtain the benchmarks reported in Figure 5. Each benchmark reports the mean runtime of 3 trials – trials were run concurrently in separate threads.

### B.6 Accuracy of CC vs Attacks

Our approaches directly leverage robust aggregation algorithms, which perform as well in our case as in the FL setting. We include an additional experiments with CC on iid EMNIST, non-iid EMNIST, and iid CIFAR100 finding that it performs as well as in the FL setting. We present the results in Table 2 and Figure 13.

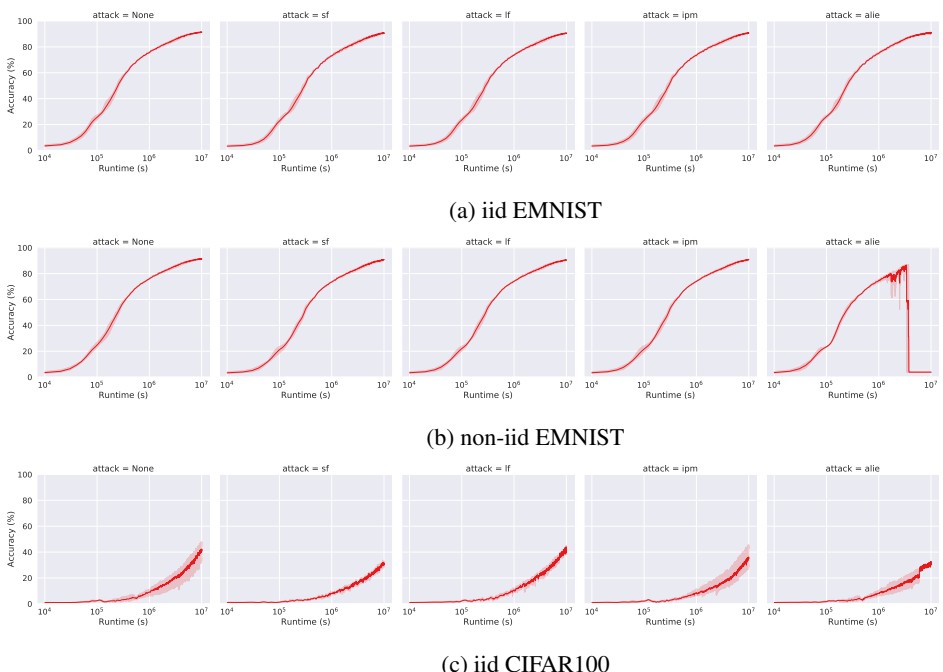

(a) iid EMNIST

(b) non-iid EMNIST

(c) iid CIFAR100

Figure 13: **The accuracy achieved by CC in a given setting**. A cohort size of 50 peers was used, with 10 malicious workers. Each algorithm was run for 1000 rounds, with 0.9 momentum, 1000.0 $\tau$, and for 3 different seeds.

