# OpenReview forum: "Robust and Actively Secure Serverless Collaborative Learning"
_NeurIPS.cc/2023/Conference — NeurIPS 2023 poster_

### Official Review · Reviewer_5nT4 · 2023-07-07

**Soundness:** 2 fair
**Presentation:** 2 fair
**Contribution:** 2 fair
**Rating:** 5
**Confidence:** 4

**Summary:**

The main contribution of this work is a decentralized learning protocol that is robust to malicious clients attempting to both hinder the learning process (model poisoning attacks) as well as break confidentiality (reconstruction attacks). Robustness against poisoning attacks is achieved by modifying a class of robust aggregation techniques that require the individual clients updates to be in a restricted space (referred to as "computational surjectivity" in the paper) and verifying the validity of the inputs (compliance with the restrictions) through distributed zero knowledge proofs (DZKPs) and using verifiable secret sharing (VSS) for aggregation. Defense against reconstruction attacks is achieved through malicious-secure multi-party computation (MPC) for aggregation. It has been claimed that all the above protocols can be implemented with reasonable efficiency for small-scale models (with up to 10^6 parameters) applied to simple problems like MNIST.

**Strengths:**

1) The paper address an important problem in collaborative learning, which is how to orchestrate collaboration when no party (clients and servers) can be fully trusted.

2) It uses a whole array of cryptographic tools (secure multi-party computation, verifiable secret sharing, distributed zero knowledge proofs) in addition to existing robust aggregation techniques to achieve the stated objectives.

**Weaknesses:**

1) First and foremost, the framing of double robustness (against malicious clients and servers) appears to be incorrect because there are no servers involved in the proposed protocol. Instead, the double robustness can be interpreted as robustness against poisoning and reconstruction attacks.

2) The most critical weakness of the proposed approach is the failure to demonstrate how all the cryptographic components will work together. While it is true that ZKPs can verify the validity of inputs, malicious-secure MPC and VSS can be used for checking the correctness of aggregation, and random committee selection can be done in a distributed fashion, it is not clear how all these tools can be tied together to form one complete system.

For example, let us consider the formation of the aggregation committee. This step itself will require a number of cryptographic operations to ensure that the committee is indeed selected "randomly" without any collusion (otherwise, the binomial approximations used to prove that the committee satisfies honest majority condition would fail).

Secondly, the honesty condition in the committee selection is merely about following the MPC protocol honestly. It does not guarantee that these committee members will not be providing malicious inputs (updates) to the collaborative learning protocol. Similarly, it does not guarantee that the committee members will verify the ZKPs honestly. There could be clients who can follow one part of the protocol honestly, but not some other part.

Finally, it is not clear if VSS, malicious-secure MPC, and DZKPs can all be implemented simultaneously. If such an implementation is possible, what are the underlying trust assumptions? Note that each of the above cryptographic primitives operate on their own trust assumptions (in terms of key generation, distribution, etc.).

3) Some of the claims in the paper appear unbelievable. For example, one of the experiments talks about having 20 peers of which there are 10 malicious workers. In this case, how can an aggregation committee with honest majority be formed using random selection? For the same experiment, it has been claimed that security does not impact robustness. Is this true even if the all the malicious workers collude with each other?

4) What is the real impact of all the robustness modifications on the final utility/accuracy? Specifically, what is the loss in accuracy compared to a vanilla FedAvg based FL aggregation by a single server? Also, what is the collaboration gain compared to stand-alone model training?





**Questions:**

1) Can you demonstrate a pipeline of how all the cryptographic primitives can be tied together to implement the complete system?  What are the trust assumptions of such a complete system?

2) Is it possible to come up with new poisoning attacks within the restrictions on the update space, possibly through collusion?

3) How will the proposed approach work in the case of real-world non-iid (hetergeneous) scenarios with distribution shifts?

**Limitations:**

No limitations have been discussed in the paper. The paper does not have any potential negative societal impact.

---

> ### Author Rebuttal · Authors · 2023-08-09
>
> Thank you for your review.
>
> >**W1. Interpretation of ‘double robustness’.**
>
> “Double robustness” communicates protection against malicious attacks mounted from the client side and server side of typical collaborative learning approaches (e.g. federated learning). We guarantee security against attacks mounted from the server side of FL by *delegating the server’s work* to a cryptographic protocol conducted by an ‘aggregation committee’ – a subset of the workers. We use ‘server’ in place of ‘aggregation committee’ in order to align with prior literature.
>
> We achieve robustness to many malicious server behaviors beyond reconstruction attacks. In most previous works a malicious server may tamper with the global model by excluding updates of certain clients, adding fabricated updates, or simply altering the global model *arbitrarily* before sending parameters to clients for the next round. **We protect against all of these attacks** and more. Specifically, clients are guaranteed to receive the correct result of the aggregation algorithm on the submitted updates.
>
> >**W2a. Security while using multiple cryptographic components**
>
> **All building blocks in our framework are secure under universal composability [UC] and thus their compositions (i.e. using them together, either in sequence or in parallel) are secure.** Concretely, in our context maliciously secure MPC protocols and DZKP all take verifiable secret shares as input; by UC these protocols can be composed together without any extra steps or concern. Key generation or distribution are not required assuming point-to-point channels, which are provided by standard TLS. *Also see answer to Q1.*
>
> [UC] Canetti, Ran. "Universally composable security: A new paradigm for cryptographic protocols." Proceedings 42nd IEEE Symposium on Foundations of Computer Science (FOCS). IEEE, 2001.
>
> >**W2b. Formation of aggregation committee.**
>
> Secure random selection of committee members is indeed required by our framework. This can be achieved using secure coin flipping, a standard and efficient cryptographic technique [CF]. We will add text to the Appendix recalling this building block (omitted for brevity).
>
> [CF] Blum, Manuel. "Coin flipping by telephone a protocol for solving impossible problems." ACM SIGACT News 15.1 (1983): 23-27.
>
> >**W2c. Security if committee members submit malicious updates or don’t follow protocol.**
>
> Our security model accounts for all of these behaviors. Submitting malicious updates (even from committee members) is tolerated via robust aggregation. VSS, malicious-secure MPC, and DZKP ensure aggregation is computed correctly – any cheating will be caught by honest committee members. Finally, the standard composition theorem [UC] applies as all building blocks are secure under composition as explained in W2a.
>
> >**W3a. Number of malicious workers in section 6.1.**
>
> Typical empirical evaluations in Byzantine robustness literature (e.g. [22]) use higher adversarial proportions than are tolerated by the cryptographic elements of our framework, and it is important to ensure that our underlying aggregation algorithms meet these standards (even when modified for efficiency). Accordingly, we benchmark *accuracy* and *robustness* of the aggregation algorithms *outside* of the cryptographic elements, finding that the robustness guarantees are retained. We will clarify that these experiments use higher adversarial proportions than are tolerated by the end-to-end protocol in the main-text.
>
> >**W3b. Security against collusion.**
>
> Our security model accounts for *arbitrary* behavior of malicious workers, including collusion.
>
> >**Q1. Pipeline of cryptographic primitives in the complete system.**
>
> We lay out the whole protocol step-by-step, elaborating on the security assumptions.
>
> 1. **Committee Election** All clients use the method from W2b to randomly select the aggregation committee $C$ with malicious security. Our analysis in Appendix C1 guarantees that $C$ has honest majority.
> 2. **Client Local Computation.** Each client computes $F^C$ and $F^P$ to obtain a preprocessed local model update given their data, and the global model parameters.
> 3. **Verifiable Secret Sharing of Updates.** Each client secret shares their update with threshold $|C| / 2$, and sends a share to each committee member. Since $C$ has honest majority, this guarantees that adversaries cannot alter or reveal the client updates.
> 4. **DZKP of valid update.** Clients prove to the committee that their updates are valid using a DZKP protocol that takes the secret shares as input. E.g. in P2P RSA, client updates must be binary-valued, so committee members create shares of a check value which is guaranteed to be $0$ if the update was binary-valued, while leaking no further information (see Appendix C.3.1 for details). Here security follows from the security of the DZKP and the VSS schemes.
> 5. **MPC for computing global updates.** Committee members compute $F^R$ in MPC, using client shares as input, to obtain a global model update. E.g. in P2P RSA, committee members sum the shares of all client updates using the standard secure addition protocol on Shamir secret shares. The committee members then reconstruct the shared sum to obtain the global update (correct reconstruction is guaranteed by VSS). As above, security follows from the security of the MPC and VSS schemes.
> 6. **Global updates sent to clients.** All committee members send the recovered value to all clients. Since $C$ has honest majority, the clients are guaranteed to recover the correct global update by accepting the majority result.
>
> >**Q2. New poisoning attacks possible within the setting**
>
> We guarantee correct computation of the underlying robust aggregation algorithm. Thus our work **only reduces** the space of possible poisoning attacks – any new attack developed in our setting would also be possible in the standard FL setting.
>
> >**Q3. Non-iid Data:**
>
> We show experiments on non-IID data (see attached pdf).

---

> > ### Comment · Reviewer_5nT4 · 2023-08-13
> > **Response to Author Rebuttal**
> >
> > I thank the authors for their detailed response, which address some of my core concerns (regarding the composition of the various cryptographic primitives and the overall pipeline). However, I would like to see a more clear presentation of the threat model and trust assumptions in the final version.

---

> > > ### Author Response · Authors · 2023-08-21
> > >
> > > Thank you for your feedback. We agree these will help the presentation and will alter the main-text Section 3 along with additions to Appendix. The main-text additions aim to clearly disambiguate the different components and highlight the key assumptions. In the appendix, we provide the full details.
> > >
> > > We made some minor alterations to the existing Section 3, and included more details at its end. The new section 3 is shown below.
> > >
> > > “
> > >
> > >
> > > Collaborative learning is conducted among a set of parties performing one of two roles: a *client* (or *worker*) who performs learning on a local repository of data, or a *server* that aggregates the many client updates. Our protocol differs in two main ways: first, it is conducted among a set of *peers* (parties) which can perform either role, and second, the role of the *server* is performed by a subset of peers termed the *aggregation committee*. To align with prior literature, we sometimes refer to peers as clients or servers when they are performing those respective roles. *We consider a malicious threat model where clients and servers may perform arbitrary adversarial actions to interfere with the protocol.* Malicious behavior in the two roles may include, but is not limited to the following.
> > >
> > > 1. **Malicious Clients** may attempt to (1) lower the quality of the trained model by sending distorted model updates. This may take the form of both (a) intentional model poisoning attacks, and (b) unintentional problems such as errors in computation, and skewed or incorrect local data sets. They may also attempt to (2) steal information about the other peers’ data, i.e. break confidentiality, e.g. by colluding with other malicious peers and sharing transcripts of the protocol execution.
> > >
> > > 2. **Malicious Servers / Committee Members** may attempt to (1) reconstruct individual data points from the clients' updates, thus breaking data confidentiality, which can be achieved by arbitrarily modifying model parameters or colluding with other parties (Committee Members or Clients), (2) inappropriately change the shared model by e.g. omitting updates from selected clients, adding in bogus updates, or otherwise altering the global model updates.
> > >
> > > We compose multiple cryptographic primitives, including secure committee election, verified secret sharing, distributed zero knowledge proofs, and secure multiparty computation. The assumptions and guarantees of the individual primitives are laid out in the Appendix C6, and their composition is secure under universal composability [UC]. Our overall protocol operates under the standard assumptions of authenticated point-to-point secure channels between peers and a bounded proportion of adversarial peers (see Appendix for details). The following are the formal guarantees of our protocol.
> > >
> > > - **Correctness of aggregation.** Given clients that submit local updates $x_1, x_2, …, x_n$, the returned global update will be equal to $F^R(x_1, x_2, …, x_n)$, where $F^R$ is a publicly known function for update aggregation. See the following section for details.
> > > - **Confidentiality of client updates.** During protocol execution, all parties gain no information about any individual client update $x_i$ beyond what is implicitly revealed by the resulting aggregation $F^R(x_1, x_2, …, x_n)$.
> > > - **Robustness to poisoning.** An accurate model will be trained even in the presence of some subset of the clients which submit poisonous updates which may take arbitrary values. Our framework compiles existing robust aggregation algorithms into a stronger security model, and thus the details of this guarantee depend on the underlying algorithm.
> > > - **Malicious security.** The above conditions hold even in the presence of a subset of parties that may perform arbitrary malicious behavior, including but not limited to collusion between malicious peers, attempts to deviate from any part of the protocol, and submission of poisonous local updates.
> > >
> > > “
> > >
> > >
> > > [UC] Canetti, Ran. "Universally composable security: A new paradigm for cryptographic protocols." Proceedings 42nd IEEE Symposium on Foundations of Computer Science (FOCS). IEEE, 2001.

---

> > > > ### Author Response · Authors · 2023-08-21
> > > >
> > > > For completeness, we show the contents of our additions to the Appendix below.
> > > >
> > > > **Byzantine Robust Aggregation.** In collaborative learning (e.g., federated learning), many clients submit model updates based on their local data. These local updates are aggregated to update the global model. In settings where clients are untrusted, some *Byzantine* or malicious clients may submit poisonous updates (which may take *arbitrary* values) with the aim of degrading the quality of the global model. Broadly speaking, Byzantine robust aggregation algorithms (often abbreviated to “robust aggregation”), guarantee that an accurate global model is trained as long as the proportion of malicious clients is bounded by a certain threshold (e.g., the theoretical analysis in CC assumes a maximum of 15% malicious clients). Further formalization of this idea occurs in a variety of ways across different works of literature – our framework is intentionally modular, inheriting the guarantees of a given underlying algorithm. Our main contribution is augmenting robustness to malicious clients with the additional guarantee that aggregation is computed *correctly and confidentially* even in the presence of malicious servers / aggregation committee members. We use the cryptographic primitives reviewed below to achieve this guarantee.
> > > >
> > > > **Committee Election.** Uniform election of committee members can be efficiently instantiated using coin-flipping [CF]. A classical way to accomplish this is to have all peers generate a string of random bits locally. The peers then make a cryptographic commitment to their random bits and distribute it to all other peers. After all peers have made their commitments, the random bits are all revealed. The concatenation of all the random bits can then be used as input to a random oracle, whose outputs can be used to select the committee members uniformly at random. This method for uniform random committee election is secure as long as at least one peer behaves honestly during the commitment process. We leverage this to guarantee that the aggregation committee has an honest majority (or is 2/3 honest in the case of FLT) (see Appendix C1 for details).
> > > >
> > > > [CF] Blum, Manuel. "Coin flipping by telephone a protocol for solving impossible problems." ACM SIGACT News 15.1 (1983): 23-27.
> > > >
> > > > **Verifiable Secret Sharing.** To make our protocols secure in the presence of malicious adversaries, we require Verifiable Secret Sharing (VSS). A VSS scheme allows the secret owner with a secret $s$, to distribute shares of $s$ among $n$ parties with a threshold $t$ such that (a) any group of $t$ parties can reveal no information about $s$ and (b) any $t+1$ parties can recover the correct value of $s$. In this work, we use Shamir secret sharing to instantiate VSS. Secrets are shared among members of the aggregation committee $C$. We make guarantees on the adversarial composition of $C$, and set $t$ such that honest parties may perform computations necessary during DZKP and MPC protocols (see below), yet adversarial parties never gain access to enough shares to reveal or modify $s$.
> > > >
> > > > **Distributed Zero Knowledge Proofs.** A malicious-secure zero knowledge proof protocol enables a prover in possession of a witness $w$ to prove to a verifier that for some publicly known function $f$, $f(w)$ takes a particular value. It is guaranteed that the verifier learns no additional information about $w$ other than what is implicitly revealed by $f(w)$, and that no malicious prover can convince the verifier that $f(w)$ takes an incorrect value. A *distributed* zero-knowledge proof (DZKP) is a variation on this primitive, wherein the prover distributes secret shares of $w$ among a set of verifiers. Leveraging the linear operations on secret shares enabled by this setting can admit particularly efficient zero-knowledge proofs (see e.g. [ZK]). In our implementations, we use DZKP protocols which assume that the set of verifiers has an honest majority.
> > > >
> > > > [ZK] Boneh, Dan, et al. "Zero-knowledge proofs on secret-shared data via fully linear PCPs." Annual International Cryptology Conference. Cham: Springer International Publishing, 2019.
> > > >
> > > > **Secure Multiparty Computation.** A malicious-secure multiparty computation (MPC) protocol enables a group of parties $P_1, P_2, …, P_n$, with respective private inputs $x_1, x_2, …, x_n$ to securely compute a function $f$ and obtain output $f(x_1, x_2, …, x_n)$. In particular, it is guaranteed that no party learns any additional information about the inputs beyond what is implicit in the output, and it is guaranteed that $f$ is computed correctly, even in the presence of parties that behave in arbitrarily malicious ways. In our implementations, we use MPC protocols which assume that the set of parties has an honest majority (2/3 honest in the case of FLT).
> > > >
> > > > *[some additional text omitted for brevity]*

---

### Official Review · Reviewer_7EiB · 2023-07-09

**Soundness:** 3 good
**Presentation:** 2 fair
**Contribution:** 3 good
**Rating:** 7
**Confidence:** 2

**Summary:**

This paper proposes a framework building on existing secure aggregation schemes to protect users from malicious server but protect the training from malicious users. If the MPC scheme used is secure then it shows that their framework is doubly robust. It illustrates this approach by leverging 3 existing algorithms (robust stochastic aggregation, centered clipping (CC), and FLTrust (FLT)) for running a grading descent on MNIST and EMNIST.

**Strengths:**

- The paper mixes several existing ideas and techniques to achieve its "doubly robust" guarantees
- The paper stays with a descent cost that allows to train a small machine learning model and reports the runtime of their experiments (fig 5)
- The paper tackles the problem of float updates and is robust to poisoning


**Weaknesses:**

- The paper seems does not mention the possibility of users to drop, which is quite common on FL. A scheme that doesn't allow it is quite unpractical
- The paper does not discuss the leakage that comes from the aggregation itself that will have the nodes. In ML we know that the aggregated gradients still leak information about individual contribution and this problem is often addressed by using Differential Privacy. Here, it seems that the authors overlook this issue.
- The paper seems does not seem primarily addressed to the ML community but rather for the security community.

**Questions:**

- could you clarify if you can handle dropout and how?
- could you discuss the risks of privacy leakage in your analysis?
- are the two curves equal in Fig 4?

**Limitations:**

The authors include a paragraph. However, it should make clearer that the security guarantees rely on the existing MPC schemes guarantees and do not provide leakage from the learnt model.

---

> ### Author Rebuttal · Authors · 2023-08-09
>
> Thank you for your review.
>
> >**User Drops:**
>
> We appreciate the question, discussion of user dropout is a great addition to our paper. We show that our framework can indeed tolerate substantial dropout. We will add the following analysis to the end of Section 4 and the Appendix.
>
> >**Tolerance of Users Dropping:** Our protocol includes two areas where peers must collaborate on the cryptographic protocol: the (client) work of computing updates and the (server / committee) work of aggregating updates. In terms of the former, our protocol tolerates any number of clients dropping, as long as the pool of clients that stays online meets the assumptions of the underlying robust aggregation algorithm. In this case the output of our protocol would be equivalent to as if only the subset of online clients submitted updates. In terms of committee member dropout, our protocols can tolerate drop out of committee members by proportionally increasing the committee size (due to the reconstruction guarantees of VSS). We analyze the committee size required to tolerate a given level of drop out in the text below, which we will add to the Appendix. We find that substantial levels of drop out can be tolerated with only modest increases to committee size. With the appropriate increases to committee size, committee member drop out (up to the specified proportion) would have no impact on the output of the protocol.
>
> >**Appendix – Tolerance of Committee Members Dropping:** In general, our protocol requires that the number of adversaries in the aggregation committee be kept below a certain proportion in order to guarantee security. The committee size is chosen as the smallest number of parties such that (except with negligible probability) a random sample from the pool of clients has less than $1/2$ adversarial proportion (in the case of RSA, CC), or less than $1/3$ (in the case of FLT). To tolerate drop out of honest committee members, we simply need to select an increased committee size such that the proportion of adversaries in the committee stays beneath these thresholds even if some number of the honest parties drop out. In particular, if we choose a committee size which guarantees (except with negligible probability) that a random sample from the pool of clients has less than $1/2 - (q/2)$ adversarial proportion, where $q$ is the proportion of tolerated dropouts from honest parties, we will guarantee that the adversarial proportion with reference to the number of committee members that stay online is at most $1/2$. We can find the necessary committee sizes by reasoning with the binomial distribution similarly to our original analysis of committee size. **For example, to tolerate 5%, 10%, and 15% dropout of honest committee members, RSA and CC would require committee sizes of 53, 60, and 69 respectively (compared to 46 with no dropout tolerance), and FLT would require 157, 218, and 326 respectively (compared to 121 with no dropout tolerance).**
>
> >**Privacy versus Security and Robustness:**
>
> We thank you for your valid concerns surrounding data privacy. We agree that our scheme provides no *data anonymization* guarantees, i.e., what can be inferred from observing the aggregated gradients, as would be protected by differential privacy (DP). We discussed this in Supplemental Appendix B and will use the extra page to move it to the main-text.
>
> Our paper provides guarantees on security (i.e. *data confidentiality*) and robustness, which are distinct from DP and of independent importance. Consider providing DP without any form of cryptographic security guarantee: this would either require local DP (which is useless in machine learning due to utility loss) or a trusted server which is often not a practical assumption and is independently vulnerable to other attacks even with DP (see our Table 1). We further remark that [7] shows that unless the server is trusted, DP is largely ineffective against reconstruction attacks whereas, as our work shows, security prevents these attacks. Though our work also provides some limited data anonymization guarantees by ensuring only the aggregated gradients are revealed, we opt to not discuss this in the paper to not create ambiguity in our contributions. We believe it is important future work to determine how to combine our work with differential privacy, which has in the past shown to require non trivial analysis [20] or an additional honest-but-curious privacy guardian [15].
>
> Finally, we note that we limit our usage of the term “privacy” in the main-text to avoid confusion surrounding cryptography and differential privacy. We hope that by moving the limitations to the main-text, this will further facilitate their distinction.
>
> >**Best Community of Interest:**
>
> Our work is a design and application of cryptography to a largely machine learning problem of “robustly learning” in the face of potentially corrupted data. We believe that our work is most interesting to those in the machine learning community because it enables this community to immediately gain security guarantees for robust machine learning.
>
> >**Question about Fig 4**
>
> Yes, the curves are roughly equal, indicating that substituting floats for fixed points does not degrade robustness.

---

> > ### Comment · Reviewer_7EiB · 2023-08-14
> > **Raising my score as authors addressed my concerns**
> >
> > I thank the authors for their clear and detailed rebuttal. I am happy to see that my question on dropouts lead authors to extend their results to take this possibility into account, which make their contribution much more practical.
> > I am also happy by the fact that the clarification on the sense of privacy will be in the main text. Finally, the authors clarified and extended the experimental part, so the contribution seems quite strong.
> > (Please change your table formatting when including the new experiments in the paper)

---

> > > ### Author Response · Authors · 2023-08-21
> > >
> > > We thank the reviewer for the especially helpful feedback and for the engagement with our rebuttal. We will indeed incorporate these suggestions into the final version of the text.

---

### Official Review · Reviewer_taqA · 2023-07-10

**Soundness:** 3 good
**Presentation:** 3 good
**Contribution:** 3 good
**Rating:** 6
**Confidence:** 2

**Summary:**

The paper proposes a generic P2P learning framework that is simultaneously secure against malicious servers and robust to malicious clients. This is achieved by combining peer-based secure aggregation with existing robust aggregation techniques in an optimized way. The P2P approach eliminates the centralized server and instead has peers that take turns aggregating the updates. This removes the power asymmetry that allows servers to breach privacy.

**Strengths:**

* The approach is shown to be computationally efficient, training models with up to 1 million parameters on standard datasets with 100s of peers.
* Strong approach towards P2P robustness


**Weaknesses:**

* The experiment on IID EMNIST is rather weak. The paper should have more convincing empirical experiments.

**Questions:**

N/A

---

> ### Author Rebuttal · Authors · 2023-08-09
>
> Thank you for your review.
>
> >**Clarification of contributions**
>
> We emphasize that the primary contributions of our work are:
>
> 1. Proposing a strengthened security model for collaborative learning that protects against malicious behavior from both clients and servers
> 2. Providing a flexible framework for realization of *existing* robust aggregation algorithms within this security model
> 3. Implementing three efficient examples of our framework applied to *existing* robust aggregation algorithms
>
> As such, the primary aim of our empirical evaluation (in particular Figures 4 and 12) is not to test the performance of the underlying robust aggregation algorithms, but rather to show that our protocols do not degrade in robustness in comparison to the centralized setting. Beyond this, the relative strength in performance of the robust aggregation algorithms (RSA, CC, FLT) is a matter of concern for those respective works. Byzantine robust aggregation is an active area of research, and our framework is intentionally modular so that new aggregation algorithms with better performance can be lifted to our security model easily.
>
> >**Additional Empirical Evaluation**
>
> Our approaches directly leverage robust aggregation algorithms, which perform as well in our case as in the FL setting. As requested, we do include an additional experiment with CC on non-iid EMNIST and iid CIFAR100 finding that it performs as well as in the FL setting. *Please also see the attached PDF to the main response with additional experimental results.*
>
> | Dataset  | Type   | No attack | sf attack | lf attack | ipm attack | alie attack |
> |----------|--------|-----------|-----------|-----------|------------|-------------|
> | EMNIST   | IID    | 91.68     | 91.09     | 90.83     | 91.2       | 91.43       |
> | EMNIST   | nonIID | 91.69     | 91.07     | 90.88     | 91.19      | 87.2        |
> | CIFAR100 | IID    | 48.04     | 33.77     | 44.94     | 45.64      | 32.77       |

---

### Official Review · Reviewer_HYce · 2023-07-27

**Soundness:** 4 excellent
**Presentation:** 4 excellent
**Contribution:** 3 good
**Rating:** 7
**Confidence:** 4

**Summary:**

Collaborative ML methods protect user data however they typically remain vulnerable to either the server or clients deviating from the protocol. Both clients and servers require a guarantee when the other cannot be trusted. This paper proposes learning scheme that
is secure against malicious servers and clients.

**Strengths:**

- The paper is tackling a very important problem in machine learning that is very relevant to the NeurIPS community.
- The paper provides the first collaborative protocol that is robust to both malicious clients and servers and operates under a malicious threat model.
- Almost any aggregation algorithm can be converted to the proposed P2P security model -- it is very flexible. The authors do this for some popular and widely used methods.
- The authors  prove the cryptographic security of their protocol.
- The presentation of the paper is very good and it is easy to follow.
- The authors perform extensive experiments demonstrating the byzantine robustness benefits and computational efficiency (and their tradeoffs) and their method performs well in all metrics.


**Weaknesses:**

This is a good paper and I think it would be of interest to the NeurIPS community. However, the idea of this paper is very simple. This is not a weakness on its own but it raises a question about the optimality of the proposed method. Although the model performs well in experiments, I am not sure whether about optimality. It would be good if the authors can comment on this.

**Questions:**

Please see weaknesses.

**Limitations:**

The limitations are addressed.

---

> ### Author Rebuttal · Authors · 2023-08-09
>
> Thank you for your positive review!
>
> >**Optimality of our Approach:**
>
> This is an interesting question. There are two axes under which we may consider optimality: robustness and security. Unfortunately, along either there exist no strong lower bounds which could be used for such an argument. We elaborate below.
>
> In the area of robustness, learning in the face of corrupted data remains an open challenge that has numerous different approaches depending on the exact threat model. This actually serves as motivation for our work which we discuss, e.g., in the conclusion—we design our protocol to be generic so as to enable broad classes of future robust aggregation algorithms to be instantiated in our protocol. Though we cannot claim or discuss optimality in terms of robustness, we hope our empirical evaluation of the protocol highlights that it is both computationally efficient and compatible with many robust aggregation algorithms, enabling the best robustness guarantees to be extended efficiently.
>
> Regarding security, we assume a security model with stronger guarantees than (to our knowledge) any previous works in collaborative learning and design a flexible framework for realizing these guarantees. The security model specifies malicious security against clients and servers, along with protection from poisoning attacks. Regarding the performance of our cryptographic protocols, although we could not argue optimality due to the lack of lower bounds, it is unlikely that the performance could be further improved significantly. We used state-of-the-art cryptographic building blocks and took full advantage of the underlying learning algorithm. However, there could be other trade-offs between security and performance: if we assume a weaker security model, e.g., assume that less number of committee members can be corrupted by a malicious adversary, further performance improvements could be explored. As such, the optimal solution depends on the exact application setting.

---

> > ### Comment · Reviewer_HYce · 2023-08-14
> >
> > I would like to thank the authors for the response. I have read the rebuttals and all the other reviews. I still believe this is a paper with solid contributions and will keep my score as is.
> >
> > I think the authors also did a very good job in the rebuttal period. I believe including some parts of the rebuttal in the final version would be very beneficial -- specially the response to Q1 by reviewer 5nT4.

---

> > > ### Author Response · Authors · 2023-08-15
> > > **Thank you for your response & recommendation**
> > >
> > > We are grateful to the reviewer for the positive evaluation and for upholding the high score for our submission. Per the suggestion to include the response provided to Question 1 from Reviewer 5nT4, we will incorporate this material in the camera-ready version of the main paper by adding the allowed extra page of content, if our work is accepted. We believe this will enable us to fully address the question in the main text, benefiting readers and improving the completeness of the work. We appreciate the reviewer taking the time to provide this recommendation to strengthen our paper, and for recognizing the value of our research contributions.

---

### Official Review · Reviewer_8qrt · 2023-07-27

**Soundness:** 4 excellent
**Presentation:** 4 excellent
**Contribution:** 4 excellent
**Rating:** 9
**Confidence:** 4

**Summary:**

The authors provide a solution for collaborative learning there are risks associated with collaboration due to clients or server(s) acting maliciously. Malicious clients can submit corrupted updates which leads to the failure of creating a useful shared model. Similarly, server can also act malicious, e.g. in data aggregation. The authors propose a Peer-to-Peer Learning that provides a doubly robust protocol against malicious clients and server(s) to train a shared model without a central party. The paper has a very nice flow, is technically sound with interesting notation and proofs (I enjoyed reading), and has experiments to back up the claims and efficiency of the proposed algorithms.

The framework is designed as a generic compiler that can efficiently convert robust aggregation algorithms to the P2P learning setting with the guaranteed malicious-secure protocol.

**Strengths:**

The paper is easy to follow. I enjoyed reading this paper. The paper presents an interesting and important setting where without a central party we want to have a doubly robust protocol against malicious clients and server(s). The contributions are solid, and worth sharing with the world. The paper is technically sound. The notations and graphs are clear, making it easy to follow the paper.

**Weaknesses:**

I do not see much

**Questions:**

My questions is related to the scalability. If a cross-device settings with pool pf millions of clients available, where we select in order of 1000-5000 clients per round in an FL round, are we looking at a million client runtime, or thousand client runtime? If it's millions, is it going to be in order of days per round?

**Limitations:**

The only weakness I can see, is the scalability of the solution and the runtime. I see runtime in order of hours for thousands of clients. Not sure how this can be scalable for very large setups. For example, a per-round CPU time of 46 seconds with 10^5 parameters (tiny model) when trained by 1000 peers shows the limitation of the scalability (and this is done on a pretty strong hardware of m5.metal of AWS. Can authors comment on this?

---

> ### Author Rebuttal · Authors · 2023-08-09
>
> Thank you for your positive review!
>
> >**Question regarding scalability when subsampling clients from a larger pool**
>
> This is a great question. To maintain security in this setting, it is necessary for the subsample of clients and selection of committee members to be known by participants in each round. This could be efficiently accomplished using standard techniques such as secure coin flipping [CF], e.g., before the protocol commences. Besides this addition, the computational cost of our protocol in this setting is the same as running it among the number of clients subsampled for a given round (in Reviewer’s example, 1000-5000 clients) for which the empirical results can be seen in Figure 5 (b). To answer the question directly, we are looking at **thousand client** runtime in this case rather than million client runtime.
>
> [CF] Blum, Manuel. "Coin flipping by telephone a protocol for solving impossible problems." ACM SIGACT News 15.1 (1983): 23-27.
>
>
> >**Limitations – Scalability**
>
> Indeed, there are still limitations that prevent full pre-training of large models. We note that this study is the first work in our security model, and further optimizations are likely possible. For example, taking advantage of parallelization, parameterizing the size of field elements used for Shamir secret sharing, and/or training using lower-precision fixed points may all provide substantial increases in efficiency. These finer-grained optimizations may be interesting to explore in future work.
>
> However, we note that the present efficiency of our scheme would enable parameter-efficient techniques [LoRA] for fine-tuning of large models, central pretraining with downstream tuning (e.g., [Gboard]), or pre-training of medium to smaller models.
>
> [LoRA] Hu, Edward J., et al. "Lora: Low-rank adaptation of large language models." arXiv preprint arXiv:2106.09685 (2021).
> [Gboard] Xu, Zheng, et al. "Federated Learning of Gboard Language Models with Differential Privacy." arXiv preprint arXiv:2305.18465 (2023).

---

> ### Comment · Reviewer_8qrt · 2023-08-10
> **Great paper, considering the rebuttal and other comments**
>
> I appreciate the complete response from the authors to respond to my concerns and other reviewer's. I read all other comments and reviewers from other reviewers, and I decide to keep my score. This is a solid and strong submission, and I think it would be beneficial for the community to hear about it.
>
> Some suggestions:
> 1. Include the following that you stated in the paper
> > To maintain security in this setting, it is necessary for the subsample of clients and selection of committee members to be known by participants in each round. This could be efficiently accomplished using standard techniques such as secure coin flipping [CF], e.g., before the protocol commences.
>
> 2. (can ignore this if you like)
> The phrase "doubly robust" is hard for someone to understand without reading the paper. I believe another simpler name could attract more people to this paper :)

---

> > ### Author Response · Authors · 2023-08-13
> > **Rebuttal, additional statement, and the title**
> >
> > We appreciate the reviewer's response, engagement in the discussion, and the prompt assessment of the rebuttal. We included the recommended statement in Section 6.2 in the main paper. Thank you also for the suggestions regarding the title - we are considering a few options, for example: "Maliciously Secure and Robust Peer-to-Peer Collaborative Learning" or "Efficient Maliciously Secure and Robust Peer Learning". We are also open to other recommendations.

---

### Author Rebuttal · Authors · 2023-08-09

We thank the reviewers for their positive feedback, insightful comments, and clarifying questions. Your reviews helped us to improve the paper. Particular thanks are due to reviewers 7EiB and 8qrt – the former facilitated an analysis of user dropout. We find that our framework can tolerate substantial dropout with only a modest increase to committee size. The latter facilitated discussion of a setting where at each round, clients are subsampled from a large user pool. We find that for little additional overhead, this setting would enable a much larger set of users to participate in training. Both discussions improve the practicality of the proposed framework.

Overall, we tackle an important problem of how to provide secure and robust collaborative learning protocols and show how to do so under the strongest setting where collaborating parties can act maliciously (reviewers 8qrt, HYce, 5nT4). This is the first framework designed as a generic compiler that can convert robust aggregation algorithms to efficient approaches in the malicious P2P learning setting (reviewers 8qrt, HYce). Our approach combines many cryptographic tools (secure multi-party computation, verifiable secret sharing, distributed zero knowledge proofs) in a generic yet tailored way to existing robust aggregation techniques to achieve the stated objectives (reviewer 5nT4). The approach is shown to be computationally efficient, training models with up to 1 million parameters on standard datasets among 100s of peers (reviewer taqA). We illustrate the approach by leveraging 3 existing robust aggregation algorithms (reviewer 7EiB). Overall, the submission ​​”has a very nice flow, is technically sound with interesting notation and proofs, and has experiments to back up the claims and efficiency of the proposed algorithms” (reviewers 8qrt, HYce).

---

### Decision · Program_Chairs · 2023-09-21

**Decision:**

Accept (poster)

**Comment:**

This work concerns settings where the servers and clients can potentially be untrusted in FL. On one hand, the server may breach the clients’ confidentiality or alter model parameters. On the other hand, the clients may perform attacks such as model poisoning. In such an environment, the paper proposes a peer-to-peer collaborative learning mechanism that is secure against malicious servers and is robust to malicious clients. The reviewer found the presentation clear and easy to follow. While simple, the idea of the paper was appreciated by the reviewers. There were some concerns about scalability and some experiments, which were addressed and discussed in the rebuttal. I also had a look at the paper and found the problem and protocol interesting to the NeurIPS community. A related line of research that can be used as motivation and in the related work section is the line of research on federated learning without a trusted server.  In addition, I encourage the authors to include their rebuttal related to the thread model, scalability/limitations, and dropouts in their final version.